# Conformational switching in the coiled-coil domains of a proteasomal ATPase regulates substrate processing

Aaron Snoberger[1], Evan J. Brettrager[1,2] & David M. Smith [1]

Protein degradation in all domains of life requires ATPases that unfold and inject proteins into compartmentalized proteolytic chambers. Proteasomal ATPases in eukaryotes and archaea contain poorly understood N-terminally conserved coiled-coil domains. In this study, we engineer disulfide crosslinks in the coiled-coils of the archaeal proteasomal ATPase (PAN) and report that its three identical coiled-coil domains can adopt three different conformations: (1) in-register and zipped, (2) in-register and partially unzipped, and (3) out-of-register. This conformational heterogeneity conflicts with PAN's symmetrical OB-coiled-coil crystal structure but resembles the conformational heterogeneity of the 26S proteasomal ATPases' coiled-coils. Furthermore, we find that one coiled-coil can be conformationally constrained even while unfolding substrates, and conformational changes in two of the coiled-coils regulate PAN switching between resting and active states. This switching functionally mimics similar states proposed for the 26S proteasome from cryo-EM. These findings thus build a mechanistic framework to understand regulation of proteasome activity.

---

[1] Department of Biochemistry, West Virginia University School of Medicine, Morgantown, WV 26506, USA. [2] Present address: Department of Pharmacology and Toxicology, University of Alabama at Birmingham, Birmingham, AL 26501, USA. Correspondence and requests for materials should be addressed to D.M.S. (email: dmsmith@hsc.wvu.edu)

Across all domains of life, the proteasome is responsible for the majority of targeted protein degradation in cells. Surprisingly, despite its crucial role in virtually every cellular process, its detailed mechanism is poorly understood. The main proteasome species in eukaryotes, the 26S proteasome, is composed of 2 subcomplexes: the 19S, which recognizes ubiquitinated substrates, and the 20S, which degrades substrates inside its hollow interior. The 19S uses ATP hydrolysis energy to unfold proteins and inject them into the 20S for degradation. 26S cryo-electron microscopic (cryo-EM) structures reveal that the 19S undergoes considerable conformational changes in response to substrate and/or ATPγS binding that appear to place the 26S proteasome into a functionally competent conformation for protein degradation[1,2]. These conformational changes seem to center around one of the 19S ATPase's coiled-coil (CC) domains (the Rpt6/3 CC). The CC domains are composed of α-helical extensions of the 19S's AAA+ ATPase subunits (Rpt1–6) that dimerize to form 3 CCs (Rpt1/2, Rpt6/3, and Rpt4/5 CCs). The CCs are intimately associated with many 19S scaffolding subunits, substrate receptors, and deubiquitinases. For example, the Rpt1/2 CC is primarily associated with Rpn1, which is a docking station that coordinates multiple ubiquitin-processing factors (e.g., the deubiquitinase USP14/Ubp6)[1,3,4]. The Rpt6/3 CC is bound to the ubiquitin receptor Rpn13 via Rpn2[1,4–14]. Additionally, after substrate binds the 26S, the Rpt4/5 CC binds the Rpn10 ubiquitin receptor as well as the proteasome's primary deubiquitinase, Rpn11[1]. Thus the CCs physically connect substrate recruitment and ubiquitin processing to the unfolding machinery. This alone indicates that CC domains play fundamental roles in proteasome function. Mutagenesis of the CCs and CC peptide competition studies further corroborate the functional importance of the CCs, since most perturbations to any CC render the 26S proteasome non-functional, leading to lethality[15,16].

Several studies suggest that posttranslational modifications (PTMs) of the CC domains affects ATPase and substrate-processing activities, indicating their importance for regulating proteasome function[17–29]. Early studies on the archaeal homolog of the 19S ATPases (proteasome-activating nucleotidase ("PAN")) also found that partial truncations of CCs affected nucleotide hydrolysis rates and even nucleotide specificity (full-length PAN hydrolyzed only ATP and CTP, whereas truncated PAN hydrolyzed ATP, CTP, ITP, GTP, TTP, and UTP)[30]. PAN's CC domains have also been shown to have chaperone activity (e.g., they prevent protein aggregation, which the 19S also exhibits)[31,32]. Though ATP hydrolysis was not required for chaperoning, ATP binding did enhance the CC's chaperone activity[30,33]. These prior studies indicate that CC domains are allosterically linked to the ATPase domains, yet no studies have shown how the CCs function mechanistically to regulate protein degradation by the proteasome. Much like the 19S ATPases, some studies suggest that the CC domains from PAN are also involved in substrate binding, although it is thought that PAN can achieve this without using additional substrate receptors that are found in the 19S[30,31,33–36].

CCs are among the most intensely studied and best understood tertiary structures and comprise two or more α-helices that wrap around one another in a knobs-into-holes fashion[37]. Dimeric, right-handed CCs (the type in proteasomal ATPases) have a repeating seven-residue consensus sequence[37,38]. By convention, these residues are named a→g, where residues "a" and "d" are hydrophobic, "e" and "g" are charged, and the rest are typically polar (Fig. 1a)[37]. Dimerization is primarily stabilized by hydrophobic interactions between the "a" and "d" residues[37,39]. While CCs are often found in fibrous proteins and other rigid structures, some CCs undergo dynamic conformational rearrangements that can regulate the protein's function[40–44]. For example, dynein's

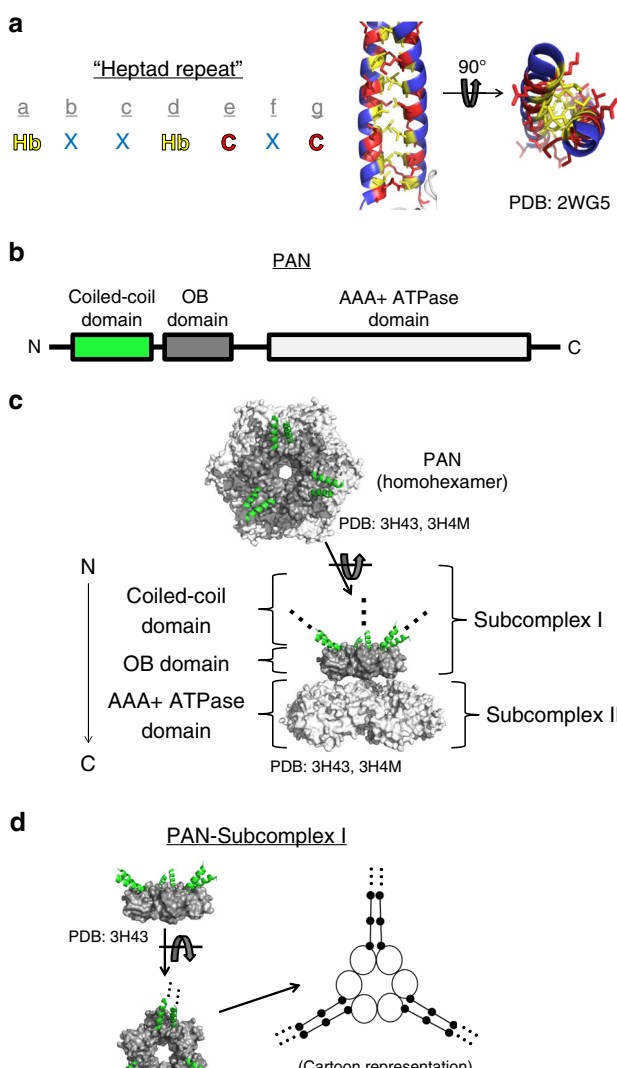

**Fig. 1** PAN contains three right-handed, dimeric coiled coils on its N-terminus. **a** Right-handed, dimeric coiled-coils (CCs) contain a seven-residue repeating consensus sequence (i.e., heptad repeat) where residues "a" and "d" are hydrophobic (Hb, yellow), "e" and "g" are charged (C, red), and the rest are typically polar (X, blue). Hb residues pack on the interior of the CC and form the main stabilizing interactions (PDB: 2WG5 [https://doi.org/10.2210/pdb2WG5/pdb]). **b** The proteasome-activating nucleotidase (PAN) from archaea contains an N-terminal CC domain, followed by an oligonucleotide/oligosaccharide binding (OB) domain and a AAA+ ATPase domain. **c** Crystal structures of the N-terminus of PAN (subcomplex I, or CC-OB domain) show PAN's six identical subunits in a symmetric ring (PDB: 3H43 [https://doi.org/10.2210/pdb3H43/pdb]). The AAA+ ATPase domain (PDB: 3H4M [https://doi.org/10.2210/pdb3H4M/pdb], light gray) also forms a hexameric ring underneath the OB domain (dark gray). Three dimeric CC domains protrude from the OB domains (green α-helices). The OB and ATPase domains domain are shown as a spacefill and the CC domains are shown as helices (PDB: 3H4M [https://doi.org/10.2210/pdb3H4M/pdb], 3H43 [https://doi.org/10.2210/pdb3H43/pdb]). **d** The structure of PAN's CC-OB domain (subcomplex I) shown from a side and top view. An arrow points to the cartoon version of the CC-OB domain complex. The backbone of the CC domains are depicted as unwound sticks, while the black dots represent the inward facing hydrophobic residues. PDB: 3H43 [https://doi.org/10.2210/pdb3H43/pdb]

CC stalk undergoes a 1/2 heptad registry shift (i.e., a shift in sequence alignment between the α-helices in a CC) to allosterically transmit long-distance signals in response to the nucleotide-bound state to modulate microtubule binding[40]. A theme among dynamic CCs is that they are rigid enough to help retain structure but flexible enough to allow movement of protein domains and/or send signals to distant domains via conformational rearrangements.

In the present study, we examine the hypothesis that proteasomal CCs must maintain not only a structure rigid enough to maintain subunit interactions but also flexible enough to permit conformational changes within the ATPase ring to allow ATP-dependent substrate translocation to occur. In addition, we hypothesize that their position above the ATPase domains and their significant integration within the 19S complex uniquely positions them to transmit allosteric signals between substrate-binding components and the ATPase complex. Since PAN effectively models the fundamental functions of proteasomal ATPases, we engineer disulfide crosslinks into the CC domains of PAN to probe the conformation of its CCs and determine whether CC dynamics or conformational changes are necessary for proper ATPase function. Surprisingly, we find that although PAN is a homohexamer, its three identical CCs individually adopt three distinct conformations. Interestingly, although these three conformations are dependent on the presence of the AAA+ ATPase domains, their general conformations do not switch during ATP hydrolysis or substrate unfolding. However, local conformational changes within two CCs are required for PAN to switch between active and resting states, and these two states can be stabilized via specific disulfide crosslinks. These conformational states in PAN, which are regulated by its CC domains, may be functionally related to activated and resting conformations previously described in 19S ATPase structures[1,5,8,9,11,13,14,45]. The allosteric linkages we establish also demystify some previously confounding observations in early studies of PAN.

## Results

**Only one of the three CCs are in the expected conformation.** PAN is a homohexamer composed of a trimer of dimers. These dimers are held together by a CC domain composed of N-terminal α-helical domains from two separate PAN subunits (Fig. 1b, c). While no high-resolution structure of full-length PAN hexamers exist, there are crystal structures of hexameric CC-OB (coiled-coil-oligonucleotide/oligosaccharide binding) domain fragment, named subcomplex I (PDB: 3H43 [https://doi.org/10.2210/pdb3H43/pdb]). Although only a small portion of the CC domain is resolved in these structures (residues 74–150 in PDB: 3H43 [https://doi.org/10.2210/pdb3H43/pdb] from *Methanococcus jannaschii* and residues 57–134 in PDB: 2WG5 [https://doi.org/10.2210/pdb2WG5/pdb] and 2WG6 [https://doi.org/10.2210/pdb2WG6/pdb] from *Archaeoglobus fulgidus*), they clearly show three CCs in-register and fully formed up to the OB domain ending at residue M87 in PAN from *M. jannaschii* (the species used for this analysis)[33,38] (Fig. 1c, d).

To determine whether CC domains in the full-length and active form of PAN are similarly in-register and zipped as observed in the crystal structure and to determine whether this conformation is functional, we started by mutating the most proximal hydrophobic "d" residue, M87, to a cysteine ("M87C" or "87C" mutation) (Fig. 2a). Because methionine and cysteine have similar stabilities in the CC "d" position, mutation alone should not significantly affect CC stability[46]. Under oxidizing conditions, disulfide crosslinks will form almost instantaneously when their β-carbon residues come between 3.4 and 4.6 Å from one another at the appropriate angle (pseusobond angles: $60° < \theta_{ij}, \theta_{ji} < 180°$,

$0° < |\theta_{ij} - \theta_{ji}| < 105°)[47]$, and cysteine residues cannot crosslink if these conditions are not met. Therefore, based on PAN's CC-OB domain structure (PDB: 3H43 [https://doi.org/10.2210/pdb3H43/pdb]) the M87C mutation should allow all CCs to form 87C-87C disulfide crosslinks under oxidizing conditions (Fig. 2a). We incubated wild-type (WT)-PAN or PAN-M87C mutants with oxidizing reagent tetrathionate (TT) or reducing agent dithiothreitol (DTT) for 1 h at room temperature and separated them via non-reducing sodium dodecyl sulfate-polyacrylamide gel electrophoresis (SDS-PAGE). Based on the crystal structure, we expected to observe only monomeric WT-PAN and only dimeric oxidized PAN-M87C. WT-PAN was monomeric as expected ($n = 16$), but we were surprised to find only $32.1 \pm 3.2\%$ (mean ± standard deviations, $n = 18$) of PAN-M87C were dimers (with the remaining 67.9% being monomeric) (Fig. 2b, Supplementary Fig. 1). We then sought to determine why PAN-M87C did not completely crosslink all subunits as initially expected. The incomplete crosslinking was not due to perturbation of PAN's hexameric quaternary structure, since the oxidized form of M87C PAN formed hexamers via native-PAGE (Supplementary Fig. 2a) and retained their normal WT-like functions, including their ability to activate gate opening in the 20S proteasome (Supplementary Fig. 2b).

Three other possibilities can explain incomplete M87C mutant crosslinking: (1) an incomplete oxidation reaction, (2) PAN hexamers exist in two conformational populations, one that is crosslinkable and another that is not, or (3) the three CC domains in PAN exist in different conformations and only one of the three (i.e., 33% of total) exists in a conformation that permits M87C crosslinking. In order to test possibility #1, we conducted a dose–response with oxidizing reagent TT to determine whether the 87C-87C crosslink was saturable. M87C crosslinking saturated at $31.6 \pm 0.8\%$ ($n = 3$) dimer formation (Fig. 2c), indicating that all cysteines within proximity to one another had successfully crosslinked, thus excluding possibility #1. Regarding possibility #2, we can expect that, if PAN exists in an equilibrium between two different conformational populations (e.g., crosslinkable and uncrosslinkable), given enough time, there should be a conversion of some, if not all, CCs to a crosslinkable state. To test possibility #2, we thus performed a crosslinking reaction at saturating levels of oxidizing reagent for up to 72 h. We observed 28.9% crosslinking after only 5 min and little additional crosslinking was observed at longer times (34.3% after 72 h; Fig. 2d). This demonstrates that PAN is not in equilibrium between two different crosslinkable/uncrosslinkable states, which rules out possibility #2. To test possibility #3 and ask whether PAN's three CC domains are restrained into three different conformations, we added denaturant (SDS) to determine whether loosening PAN's quaternary structures could increase the amount of crosslinking. We found that increasing denaturant concentrations resulted in an increase in 87C-87C crosslinking up to $96.3 \pm 4.9\%$ dimers ($n = 3$; Fig. 2e). This suggests that lack of 87C-87C crosslinking in ~67% of PAN-M87C's CC domains is attributed to conformational restraints preventing M87C residues from coming into close proximity.

Taken together, we conclude that the $32.1 \pm 3.2\%$ crosslinking observed in PAN's M87C residue is due to a single CC adopting a conformation similar to that found in the CC-OB domain crystal structure, while the other two CCs cannot access this same conformational state and thus 87C-87C crosslinking does not occur in these CCs. These data therefore indicate that although PAN has 3 identical CC domains, they adopt three non-identical conformations under these experimental conditions. Because these disulfide crosslinking experiments (which were done on full-length PAN) did not agree with the CC-OB domain crystal structure, we hypothesized that we could achieve 100% 87C-87C

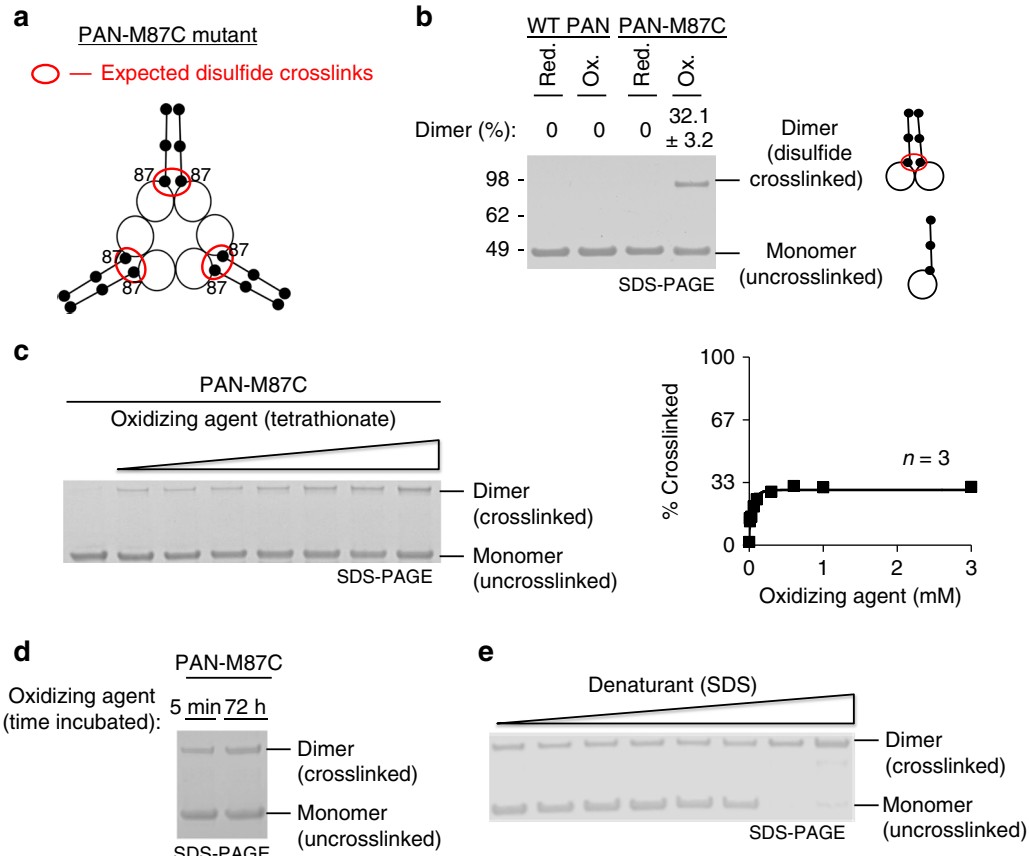

**Fig. 2** PAN's coiled-coils do not adopt symmetrical conformations. **a** Cartoon representation of the PAN-M87C mutant CC-OB domains (based on PDB: 3H43 [https://doi.org/10.2210/pdb3H43/pdb]). The PAN-M87C mutant contains a cysteine in place of methionine at the CC domains most proximal hydrophobic residue. These cysteines are expected to form a disulfide crosslink based on measured β-carbon distances in its crystal structure (PDB: 3H43 [https://doi.org/10.2210/pdb3H43/pdb]). **b** Representative non-reducing SDS-PAGE and Coomassie staining of WT-PAN and PAN-M87C under reducing (1 mM DTT) and oxidizing (1 mM tetrathionate) conditions followed by desalting (See Methods for details). The mean percentage of dimer with standard deviations is indicated at the top of the gel (WT-PAN-red- $n = 3$; WT-PAN-ox- $n = 16$; PAN-M87C-red- $n = 3$; PAN-M87C-ox- $n = 18$). See Supplementary Fig. 1 for full-length lanes of oxidized samples used for this quantification. **c** Experiment with PAN-M87C similar to (**b**) but with a dose response with oxidizing reagent (tetrathionate) prior to desalting and SDS-PAGE. The quantification of percentage of crosslinked vs. the concentration of tetrathionate is also shown on the right. Error bars are shown ($n = 3$) but are smaller than the data points. **d** Representative gel showing the amount of PAN-M87C crosslinking after 5 min or 72 h of incubation with tetrothionate ($n = 3$). **e** PAN-M87C was incubated in 1 mM tetrathionate, and increasing amounts of denaturant (SDS: 0.00006, 0.0006, 0.003, 0.006, 0.03, 0.06, 0.3, and 0.6%) were added prior to desalting and SDS-PAGE. At higher levels of SDS, >95% crosslinking was observed. A representative gel is shown ($n = 3$). See Supplementary Fig. 8 for validation of quantitative SDS-PAGE analysis of PAN

crosslinking after separating the ATPase and CC-OB domains. The first CC-OB domain crystal structure was generated via partial proteolysis to remove the ATPase domain. Therefore, we subjected our PAN-M87C mutant to similar partial proteolysis conditions (Fig. 3a) and purified the two PAN subcomplexes as previously described (Fig. 3b)[38]. We analyzed both subcomplexes via SDS-PAGE and found that subcomplex I (F3) contained an ~8 kDa fragment (the expected single CC-OB domain size) and a ~16 kDa fragment (the expected CC-OB dimer size) (Fig. 3c), while subcomplex II (F2) predominantly contained a ~30–35 kDa fragment, the expected single AAA+ ATPase domain size (Supplementary Fig. 3). We further analyzed full-length PAN (F1) and subcomplex I (F3) since these fragments contain a CC domain (whereas subcomplex II contains only the AAA+ ATPase domain)[38]. Partial proteolysis resulted in a high background, but the full-length (or near full length) fraction (F1) still clearly showed PAN dimers and monomers (Fig. 3c, F1—top of gel oxidized lane) just as observed in Fig. 2b. However, ~100% of the CC-OB domain fragment (Fig. 3c, F3—bottom of oxidized lane) was crosslinked and ran as dimers under oxidizing conditions.

Even a large fraction of the reduced CC-OB fragment crosslinked, likely in the gel, due to the necessary non-reducing conditions (Fig. 3c, F3—bottom of reduced lane). This demonstrates that, upon removing the AAA+ ATPase domains, all CC-OB domains are able to fully form the 87C-87C crosslink, consistent with this fragment's crystal structures. Since two independent methods that disrupt conformational restraints (SDS and partial proteolysis) both allow full crosslinking, we conclude that in full-length PAN, only ~1/3 of PAN's CC domains are crosslinkable because only 1 of its 3 CCs are zipped and in-register at this proximal position (implying PAN's other 2 CCs must adopt different conformations, discussed in detail below). We will call the in-register and zipped CC "C1" (for CC conformation 1).

**Nucleotides have little effect on CC conformation.** All above crosslinking experiments were performed without nucleotides. Given that the AAA+ ATPase domains seem to allosterically restrict PAN's CC domain conformations, we questioned whether the nucleotide-bound state could regulate PAN's CC domains. In

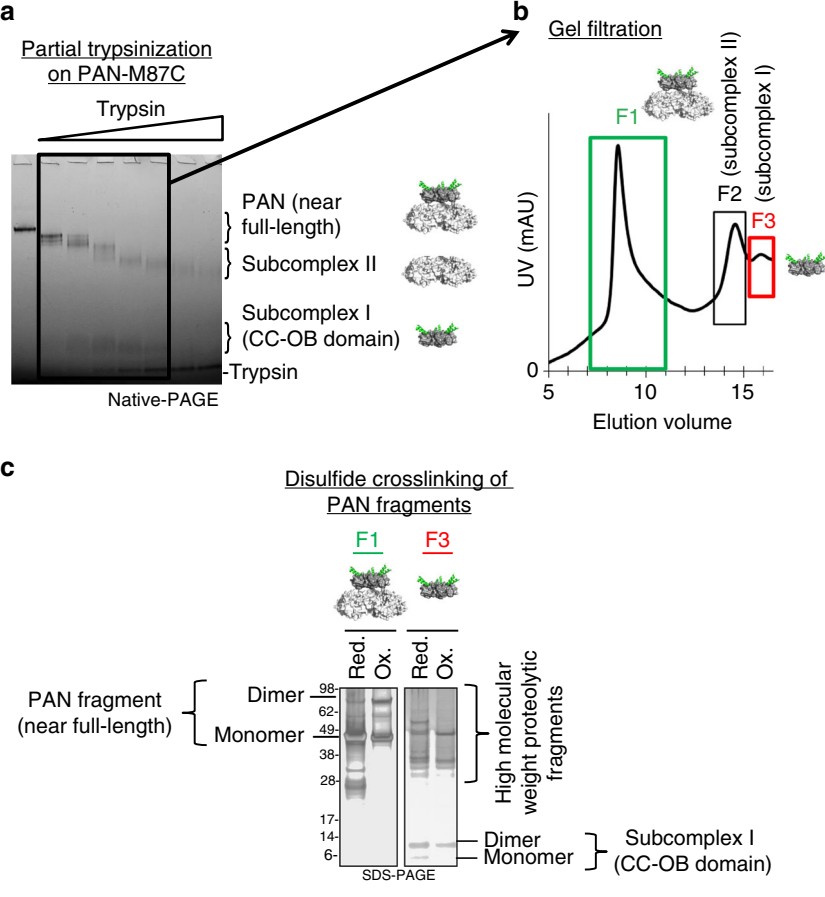

**Fig. 3** PAN's ATPase domains induce a conformational asymmetry in its coiled-coils. **a** PAN M87C was incubated with increasing amounts of trypsin and analyzed by native PAGE to generate two subcomplexes of PAN[38]. Black box indicates fractions that were pooled for gel filtration in **b**. **b** Gel filtration was conducted on pooled fractions from **a**. Three main peaks were observed that correspond to molecular weights of near-full-length PAN, subcomplex II (AAA+ ATPase domain), and subcomplex I (CC-OB domain). **c** Fractions 1 and 3 from (**b**) were incubated under reducing (1 mM DTT) or oxidizing conditions (1 mM tetrathionate), desalted, and analyzed by SDS-PAGE and silver staining

order to test this, we performed crosslinking reactions for 1 h at room temperature in the Apo (no nucleotide), ADP, ATP, and ATPγS state (Supplementary Fig. 4a). In addition, we also confirmed, via mass spectrometry (MS), that under oxidizing conditions, an 87C-87C crosslink occurred in the Apo state (Supplementary Fig. 4b). No difference in crosslinking was observed when nucleotides were bound, except for a modest decrease in crosslinking in the high ATPγS state (Supplementary Fig. 4a), which was previously shown to force PAN into an unnatural 4-ATP-bound state with suboptimal activity[48]. It is important to note that, after 1 h at room temperature, reactions are expected to have gone to completion, so these experiments could not resolve whether nucleotide binding caused differences in PAN-M87C's time to crosslinking. Therefore, we analyzed the timecourse of PAN-M87C crosslinking immediately after addition of oxidizing agent. Since crosslinking occurs very quickly, we conducted this experiment at −17 °C to attempt to resolve and compare time to crosslinking with various nucleotides added (see Methods for details). The rate of M87C crosslink formation with 2 μM ATPγS was not discernably different from crosslink formation without nucleotides. However, we did observe a slight reduction in crosslinking rate with 2 mM ADP (Supplementary Fig. 4c). Therefore, saturating ADP levels appear to slightly disorganize C1 (i.e., slows the M87C crosslinking rate), but ADP binding does not completely restrict PAN from entering the C1 conformation.

**The CCs in PAN adopt distinct conformations.** Since PAN is homohexameric, it was surprising to find that PAN's 3 CC domains are not in the same conformation. This indicates that PAN's other two CCs must be in different conformational states, perhaps unzipped or out-of-register. To test this possibility, we engineered other cysteine mutants in PAN that could allow crosslinking of other potential CC conformational states. PAN's CCs extend from residue 87 to ~50 (with 90% confidence) (Fig. 4a). In order to determine whether PAN's CCs may be unzipped proximally and then rezipped distally (see below, Fig. 4c), we systematically mutated each individual hydrophobic "a" and "d" CC residue to a cysteine starting from the proximal end and moving to the distal end. As with other CC-containing proteins, PAN's "a" residues mutated to cysteine did not easily form disulfide bonds (Supplementary Fig. 5a), likely due to unfavorable Cα-Cβ bond angles of the "a" residues[49]. Thus we limited our analysis to hydrophobic "d" residues in PAN. Using the same protocols we used for the M87C experiments, we observed a gradual increase in disulfide crosslinking formation from proximal to distal CC residues (Fig. 4b) with the 59C mutation (fifth heptad) resulting in 69 ± 5.9% (n = 14) crosslinking (59C-59C crosslink) (Fig. 4b). Based on Fig. 3, we expect that all 5 heptads in C1 (crosslinked by M87C) are in-register and zipped and thus contribute ~33% crosslinking to these experiments, so every "d" residue cysteine mutation should also crosslink in the C1 CC (Fig. 4c). Therefore, the additional

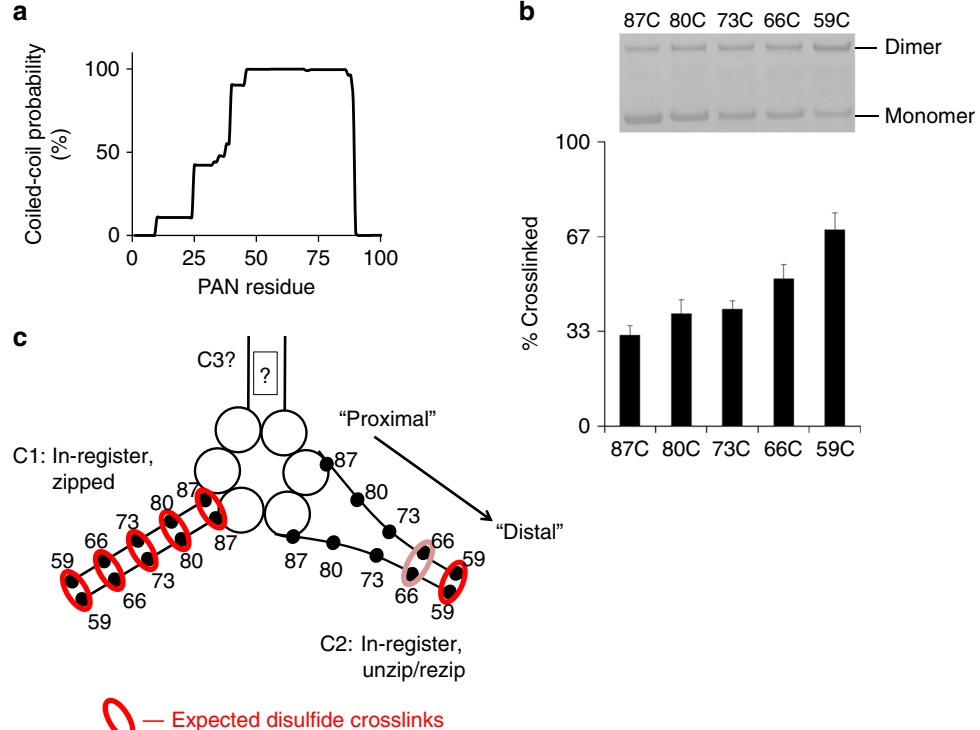

**Fig. 4** One of PAN's three coiled-coils is partially unzipped. **a** Prediction of PAN's CC domains via the Paircoil2 software[58]. **b** Oxidation followed by desalting and SDS-PAGE of the indicated "d" residue point mutants of PAN along the length of the CC. In this and following figures, we indicate the cysteine mutations by listing the residue number followed by a "C", (e.g., residue M87 mutated to cysteine is denoted as "87C"). Quantification of percentage of crosslinked is shown below in bar graph form as means ± standard deviations, (87C- $n = 18$; 80C- $n = 7$; 73C- $n = 12$; 66C- $n = 7$; 59C- $n = 14$). See Supplementary Fig. 1 for full-length SDS-PAGE lanes used for quantifications of PAN-87C and PAN-59C mutants. *P*-values for all points relative to 87C were <0.001. **c** Summary model of in-register disulfide crosslinks that can occur in PAN's C1 and C2 CCs. Red circles represent crosslinked residues, pink circles indicate partial crosslinking

crosslinking that is observed (up to ~69% in more distal cysteine mutants) likely comes from a second in-register CC that can crosslink distally but not proximally, we call this CC conformation "C2." Thus the percentage of crosslinked PAN changes from 1/3 (87C-87C crosslink) to 2/3 at 59C (59C-59C crosslinks) (Fig. 4b, c). It therefore appears that C1 is in-register and fully zipped while C2 is in-register but unzipped at its proximal residues (Fig. 4c).

Since we never found a single cysteine mutation (which probes for in-register CCs only) in PAN that could simultaneously crosslink all three of its CCs, we next sought to determine whether the third CC domain was in an out-of-register conformation or potentially unstructured. We therefore systematically generated double mutants that could crosslink both in-register and out-of-register CC conformations of different sized registry shifts (e.g., see Fig. 5a). To analyze whether PAN's third CC conformation is slidden by <1 heptad, we generated double cysteine mutations with residue 87 (the first hydrophobic "d" residue) plus one of each residue in the first heptad (86→81C) (Supplementary Fig. 5b). Such mutants can allow for crosslinking of both in-register and out-of-register conformations (e.g. an 87 +86C mutant could allow for an in-register 87C-87C and 86C-86C crosslink or it could allow for an out-of-register 87C-86C crosslink). Therefore, if a registry shift is present, we expect approximately 2/3 crosslinking under oxidizing conditions (1/3 from C1 and 1/3 from C3). However, none of these mutations had significant crosslinking above their single mutant controls (e.g., the 87+86C mutant did not crosslink more than the single 87C and 86C mutants combined)

(Supplementary Fig. 5c). This result ruled out a registry shift <1 heptad.

To test the possibility of register shifts >1 heptad, we generated cysteine mutants that could crosslink under conditions of 1-, 2-, 3-, or 4-heptad slides (Fig. 5a). Interestingly, double mutations engineered into the first 3 heptads that would allow crosslinking of a single full heptad slide (i.e., 1-heptad slide mutants; Fig. 5a) all resulted in less crosslinking than their single mutant controls (Fig. 5b), suggesting that mutating >2 consecutive "d" residues may disrupt local CC stability. However, if a 1-heptad slide did occur (even briefly), we expect that disulfide crosslinking would have stabilized this 1-heptad slidden conformation. So, although unlikely, we cannot completely rule out that PAN's third CC adopts a 1-heptad slide, since controls produce less than the expected crosslinking in these double mutants. We next engineered mutants that would stabilize full heptad slides of 2, 3, or 4 heptads (Fig. 5a—right 3 panels). Mutants engineered to crosslink 2-heptad slides did crosslink more than their single mutant controls, (Fig. 5c—left two gels), but mutants engineered to detect 3-heptad (80+59C) and 4-heptad (87+59C) slides did not show additional crosslinking (Fig. 5c—right two gels) ruling out 3-heptad and 4-heptad registry shifts. Since crosslinking of PAN (87+73C) resulted in additional crosslinking over their single mutant controls (66.5 ± 4.3% total crosslinking, $n = 7$) (Fig. 5c), this indicates a 2-heptad slide could be detected in this mutant. In the 73C single mutant, ~33% crosslinking is expected to arise from C1 (73C-73C crosslink), with ~7% crosslinking coming from C2 (Fig. 4b). We thus hypothesized that, in the 87+73C double cysteine mutant, somewhere between 27 and 34% crosslinking is contributed by the 2 heptad out-of-register CC due

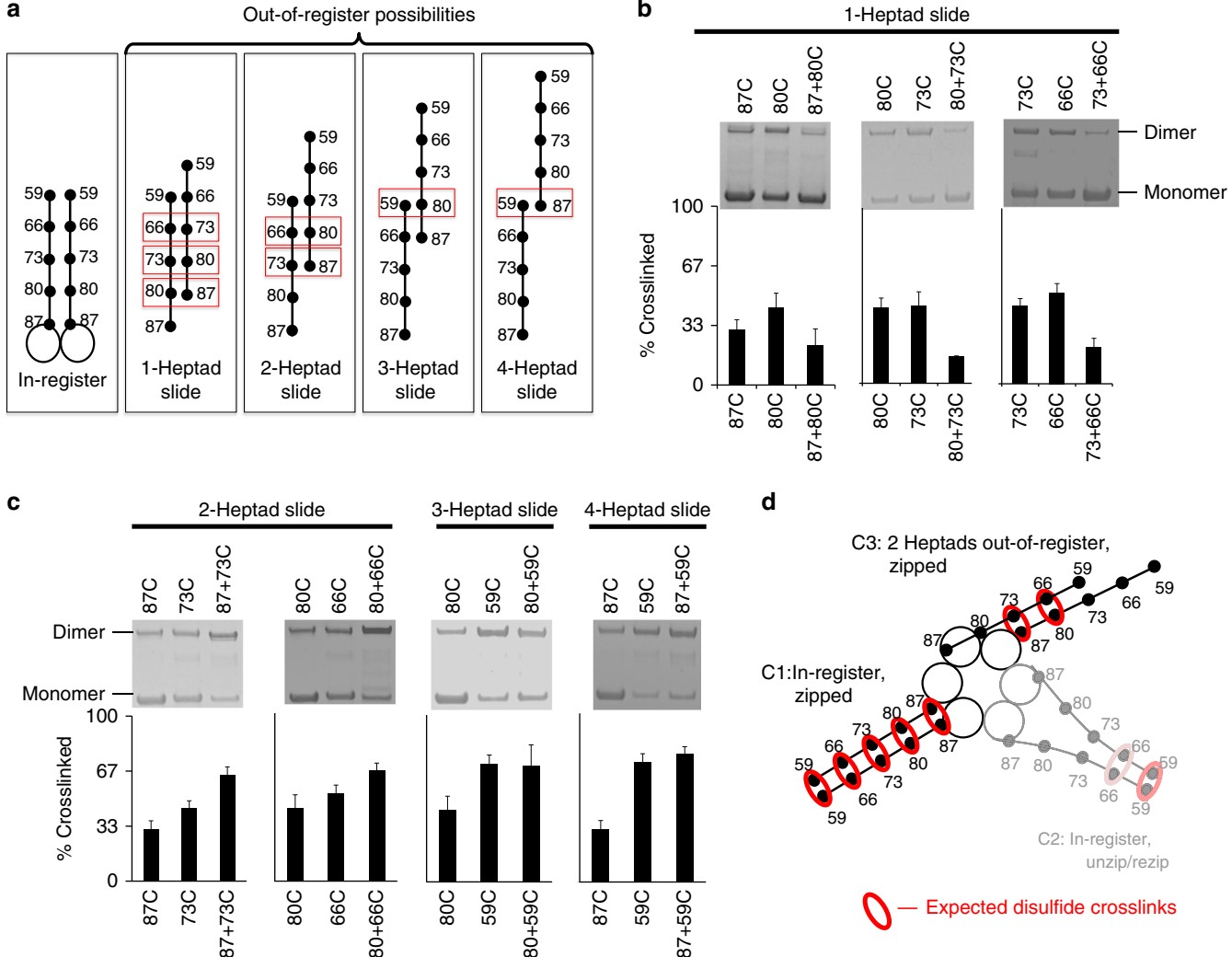

**Fig. 5** One of PAN's coiled-coils is 2 heptads out-of-register. **a** Illustration of possible CC register shifts. The boxed areas are expected crosslinks between "d" residues mutated to cysteines, assuming full zipping at these residues. **b** The indicated PAN mutants that allow for crosslinking of 1-heptad slide (double mutants) and their single mutant controls were subjected to crosslinking (1 mM tetrathionate) and run on SDS-PAGE for analysis. Percentage of crosslinked is quantified in bar graph below. **c** Same as **b** but using the indicated mutants allowing for crosslinking of 2, 3, and 4 heptad slides. Only 2-heptad slides show additional crosslinking in their double mutant compared to single mutant controls. See Supplementary Fig. 1 for full-length gels of 87C, 59C, and 87+73C used for quantifications. **d** Summary model of the crosslinks that can occur in PAN's C1 and C3 CCs (the crosslinks that form in the C2 CC are also lightly pictured here). Red circles represent crosslinked residues. Bar graphs are means ± standard deviations (87C- $n = 18$; 80C- $n = 7$; 73C- $n = 12$; 66C- $n = 7$; 59C- $n = 14$; 87+80C- $n = 6$; 80+73C- $n = 2$; 73+66C- $n = 4$; 87+73C- $n = 7$; 80+66C- $n = 4$; 80+59C- $n = 4$; 87+59C- $n = 6$)

to an 87C-73C crosslink (called conformation "C3"). To further support this hypothesis, we engineered a second mutation capable of capturing a 2-heptad slide, "80+66C", which is identical to 87+73C but 1 heptad more distally located (forming an 80C-66C crosslink). We observed 72% crosslinking, consistent with crosslinking levels observed in 87+73C (Fig. 5c), thus confirming the 2-heptad slide with a different double mutant. So, we have generated two crosslinkable mutants (87+73C and 80+66C) capable of simultaneously crosslinking the C1 (87C-87C/73C-73C and 80C-80C/66C-66C crosslinks) and the C3 (87C-73C and 80C-66C crosslinks). Both mutants were hexameric by native-PAGE (Supplementary Fig. 2a) and had similar 20S gate-opening capacity (Supplementary Fig. 2b), meaning these mutants are functional and have the expected quaternary structure. These data indicate that one of PAN's CCs natively adopts a full 2-heptad slide (C3 conformation). To our knowledge, in all other proteins where the structure of out-of-register CCs are available, sliding only

occurs over ~1/2 heptad (~4–5 residue slides), and to date, such an extensive 2-heptad registry shift has not been reported[40–43].

**CCs regulate ATPase rate and substrate processing.** The above results indicate that PAN's AAA+ ATPase domains impose structural restraints on PAN's three CC domains such that they adopt at least three distinct conformations: C1 (in-register, zipped), C2, (partially unzipped), and C3 (2 heptads out-of-register). We hypothesized that this ATPase to CC domain allosteric communication should also work in the reverse direction. In other words, CC domains, which are associated with substrate binding, may allosterically regulate the activity of the distant ATPase domains. To test this hypothesis, we sought to determine ATP hydrolysis kinetics of WT-PAN and cysteine mutants that crosslink the various combinations of C1, C2, and C3 conformations we have identified. Under reducing conditions, WT-PAN performed the same as previous observations and oxidizing

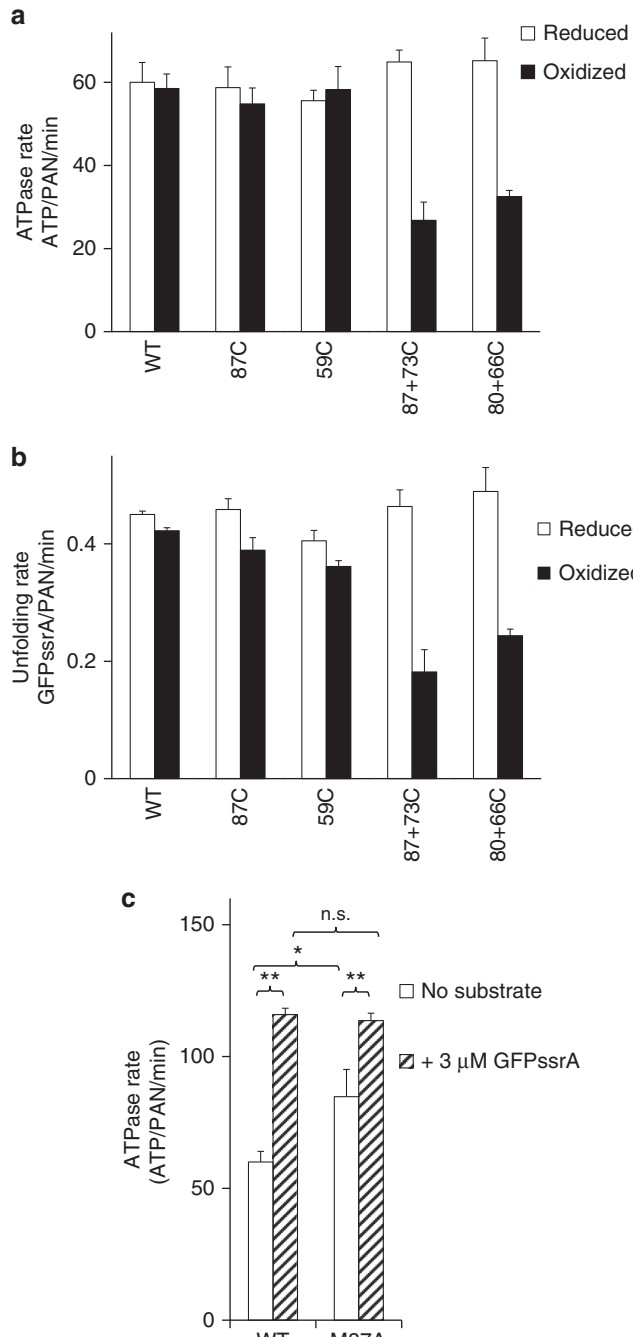

**Fig. 6** Some coiled-coil conformations regulate PAN's activity. **a** ATPase activity of the indicated PAN mutants under reducing and oxidizing conditions with saturating 2 mM ATP. ATPase activity is determined using an ATP regenerating system coupled to NADH, whose absorbance is monitored in real time. **b** Unfoldase activity of the indicated PAN mutants was measured under reducing and oxidizing conditions using saturating substrate (GFPssrA, 3 μM) with saturating ATP (2 mM). The rate of GFP unfolding was determined by assaying its loss of fluorescence in the presence of the indicated PAN and the 20S proteasome. **c** ATPase activity of WT-PAN and PAN-M87A in the absence or presence of saturating (3 μM) substrate. Bar graphs values are means ± standard deviations of three independent experiments ($n = 3$). *$P < 0.05$, **$P < 0.001$, n.s. not significant (unpaired Student's $T$-test, Sigmaplot)

conditions did not significantly alter its kinetics (see Supplementary Table 1 for values, Fig. 6a and Supplementary Fig. 6). We then determined ATP hydrolysis rates for the CC mutants in the reduced (uncrosslinked) and oxidized (crosslinked) conditions. Under reducing conditions, all mutants hydrolyzed ATP with rates similar to WT (Fig. 6a—white bars). Interestingly, under oxidizing conditions, 87C (crosslinked C1) showed normal ATPase activity and 59C (crosslinked C1 and C2) had a 25% increase in its Vmax for ATPase activity compared to its reduced control after normalizing to WT-PAN ($P < 0.001$, unpaired Student's $T$-test, Sigmaplot; $n = 3$; Supplementary Table 2). The lack of functional impact from crosslinking was unexpected, so we also verified by SDS-PAGE that the anticipated crosslinks were indeed present at the expected levels from the PAN that was previously used in the same rate reactions. Moreover, we also repeated this same experiment while monitoring PAN's ability to unfold GFPssrA (Fig. 6b) and we found similar results as when we followed ATPase activity: PAN functioned well even with crosslinked C1 and C2 conformations. While the 59C-59C crosslink covalently locks the C1 and C2 CCs, this crosslink is located toward the N-terminal end of the CC domain, which could therefore still allow for conformational changes to occur in the C-terminal side of the CC, which is located adjacent to the OB domain. However, the 87C-87C crosslink is on the most C-terminal residue in PAN's CC and stabilizes the CC domain directly adjacent to the OB domain. The fact that PAN's activity is unaltered in the 87C-87C crosslink demonstrates that PAN's C1 CC can remain in-register and zipped throughout the entire ATP hydrolysis and substrate unfolding cycle. To further test this possibility, we mutated the M87 residue to an alanine (M87A), which is a conserved mutation but is less stable at this "d" position than a methionine or cysteine (i.e., M87A will slightly and locally destabilize C1)[46]. It is known that WT-PAN has repressed ATPase activity in the absence of substrate and that addition of saturating substrate causes activation of PAN's ATPase activity. Intriguingly, the repressed state (no substrate) of PAN-M87A hydrolyzed ATP faster than WT (Fig. 6c—white bars) but had the same ATPase activity as WT in the activated state (substrate-bound) (Fig. 6c—hatched bars). Therefore, a stable C1 at the M87 residue is necessary to properly repress PAN in the absence of substrate, thus allowing substrate-mediated ATPase activation to occur.

To test the functional effects of crosslinking C3, we also determined ATP hydrolysis rates for the 87+73C and 80+66C mutants, which both crosslink C1 and C3. Interestingly, under only oxidizing conditions, we observed a substantial reduction in PAN's ATPase activity in both of these crosslinked variants (Fig. 6a). Similar results were also found when we performed the same experiments but assayed protein unfolding activity (Fig. 6b). Since crosslinking C1 has no effect on activity, and since crosslinking C1+C3 lowers PAN's activity, these data indicate that stabilizing C3 by crosslinking stabilizes a functional but low activity state of PAN. This suggests that crosslinking the C3 state results in the slowing of some step in the ATP hydrolysis cycle (ATP binding, ATP hydrolysis/phosphate leaving, or ADP off rate) but does not have an effect on the mechanochemical coupling of ATP hydrolysis to substrate unfolding previously observed for proteasomal ATPases[50]. While PAN 87+73C alone in the reduced state had a slightly lower Vmax ($42 \pm 1$ ATP/PAN/min), it had a similar Km as WT-PAN ($556 \pm 29$ mM) (Supplementary Table 1, Supplementary Fig. 6). However, when this 87+73C mutant was crosslinked under oxidizing conditions, we observed such a severe impairment in ATP hydrolysis that we could not achieve saturation with ATP, indicating a large increase in the Km for ATP hydrolysis (>3000 μM) (Supplementary Table 1, Supplementary Fig. 6). We also observed a 63%

reduction in ATPase and a 64% reduction in unfoldase activity in the other 2-heptad slidden mutant (80+66C) at high ATP (Fig. 6a, b). This demonstrates that trapping one of PAN's CCs in the C3 conformation dramatically alters ATP hydrolysis kinetics, most likely due to a specific allosteric mechanism that is regulated by the C3 CC conformation.

**CCs toggle between activated and resting states.** Given that our 87C, 59C, and 87+73C mutants each can crosslink all three of the CCs in some combination, we expected that an 87+73+59C triple mutation would crosslink all three CCs at once. However, after generating and exposing this triple PAN mutant to oxidizing conditions, we only observed $74 \pm 3\%$ crosslinking ($n = 4$) (Fig. 7a, Supplementary Fig. 1), which is consistent with only 2 of the 3 CCs being crosslinked. But which of these two CCs are crosslinked? Upon further analysis, we found that this triple mutant functions in a similar fashion as WT-PAN when uncrosslinked or crosslinked in both ATPase (Fig. 7b, Supplementary Fig. 6; Supplementary Table 1) and unfoldase (Fig. 7c) activity. The normal function of the triple mutant indicates that C1 and C2 most likely crosslink (87C-87C and 59C-59C) and the C3 (87C-73C) does not, since if C3 crosslinked one would expect a substantial decrease in both ATPase and unfoldase (as observed in PAN 87+73C, Fig. 6a, b). This demonstrates that C2 and C3

conformations cannot be simultaneously crosslinked, because only two of the three possible crosslinks could simultaneously form in this triple mutant. Based on these data, we hypothesize that there are two conformational states of PAN, state #1 where C1 and C3 are crosslinkable (which 87+73C mutant crosslinking can capture), and state #2 where C1 and C2 are crosslinkable (which the 59C mutation can capture) (Fig. 7d). State #1 appears to exist in a resting state while state #2 appears to have a slightly activated ATPase activity (~25% above its reduced control, Supplementary Table 2). We also found that the triple mutation (87 +73+59C), which could potentially crosslink either state #1 or state #2, preferentially crosslinked state #2 as it had a more similar functional profile to state #2 (59C) than to state #1 (87 +73C) (Supplementary Table 1, Figs. 6a, b and 7a, c, Supplementary Fig. 6). This indicates that PAN crosslinks more easily in state #2, which prevents switching to state #1. However, if state #2 is not stabilized via crosslinking then state #1 can be captured with the 87+73C mutant. This regulation of state changes in the ATPase domains (regulated by CC conformation) seems to be a theme in the proteasomal ATPases and these results for PAN have parallels with state changes in the eukaryotic 26S proteasome (discussed below).

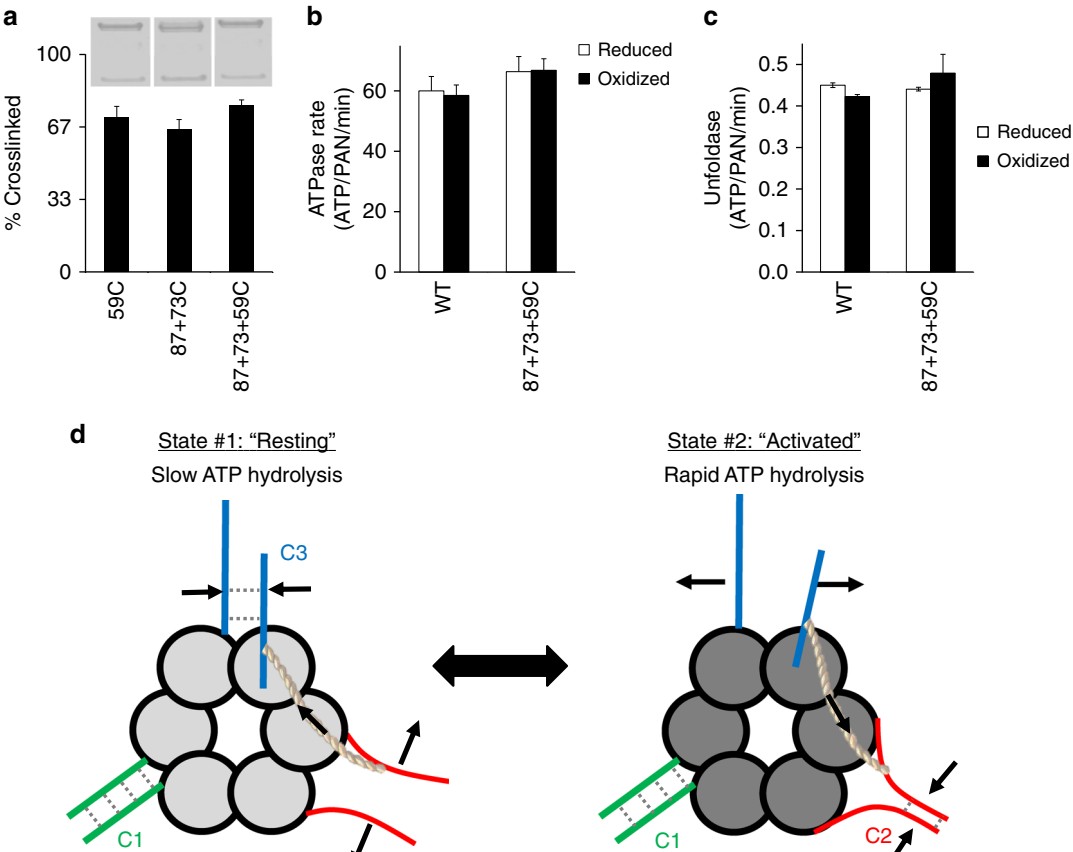

**Fig. 7** PAN adopts two distinct states: one that is fully active and another that is resting. **a** The PAN triple mutant (PAN 87C+73C+59C) was subjected to oxidizing conditions and compared to PAN 59C and PAN 87C+73C. Crosslinking experiments were performed and representative images are from a noncontiguous gel. Bar graphs are means ± standard deviations (59C- $n = 14$, 87+73C- $n = 7$; 87+73+59C- $n = 4$), See Supplementary Fig. 1 for full-length SDS-PAGE lanes of all lanes used for this quantification. ATPase activity (**b**) and unfoldase activity (**c**) of PAN 87C+73C+59C under reduced and oxidized conditions was determined and compared to WT. Bar graph values are means ± standard deviations ($n = 3$). **d** Model representing the conformational asymmetry of PAN's CC domains and how conformational switching in the CC domains regulates its activity. In state #1, PAN is in a resting state with slowed ATP hydrolysis and substrate processing, and C1 (green) and C3 (blue) CCs are crosslinkable. However, owing to allosteric restrictions (indicated by the rope linking the blue and red CCs), C2 (red) is not crosslinkable in state #1. In state #2, PAN is fully active with rapid ATP hydrolysis and substrate processing, and the C1 (green) and C2 (red) CCs are crosslinkable, while the C3 CC (blue) is not crosslinkable

## Discussion

This study uncovered several surprising properties of the structure and function of the PAN proteasomal ATPase: (1) PAN's CCs are not conformationally symmetric, (2) global conformation of the CCs does not appear to cycle around the ring, and (3) local changes in two of the CC conformations (C2 and C3) can regulate PAN's activity. Previous biochemical studies suggest that PAN adopts asymmetrical conformations[48,50,51], but the present study provides direct structural evidence of this, at least for the CC domains. This finding is critical, because up until this point it has been assumed based on crystal structures of PAN's N-terminus that it is symmetric. Interestingly, this CC conformational asymmetry in the homohexamer PAN also extends to the eukaryotic 26S proteasomal ATPases (Rpt1–6). Although CC conformational asymmetries in Rpt1–6 have not been previously discussed, it is clear from cryo-EM structures that the CC domains are also asymmetric, though this is less surprising since Rpt1–6 is a heterohexamer. In fact, one structure of the 19S ATPase's CC domains (PDB: 4CR4 [https://doi.org/10.2210/pdb4CR4/pdb]) shows its Rpt4/5 CC in a mostly zipped conformation (similar to the C1 conformation found in PAN), while the Rpt6/3 CC is partially unzipped (similar to the C2 conformation in PAN), and the Rpt1/2 CC is 2 heptads out-of-register (similar to the C3 conformation in PAN) (Supplementary Fig. 7)[5]. We measured β-carbon distances and angles between each hydrophobic residue in the Rpt CCs and found that if similar cysteine mutations were introduced in this structure, the Rpt 4/5 (C1) and Rpt 6/3 (C2) conformations would crosslink but Rpt 1/2 (C3) would not, consistent with the results we found here for PAN (e.g., 59C). Therefore, our findings with disulfide crosslinking of PAN's CC domains are fairly consistent with the CC domain conformations in the 26S. This suggests that CC conformational asymmetry is inherent to the proteasomal ATPase rings and existed prior to the development of eukaryotes. This suggests that the AAA+ ATPase domains of PAN and the 19S likely hexamerize in a similar conformational fashion (e.g., a lockwasher), which provides a structural explanation of previous observations that the 19S and PAN share similar nucleotide binding, hydrolysis, and substrate-processing characteristics where both PAN and the 19S ATPases (1) bind to 2 ATPs[48], (2) bind ATPs on adjacent subunits[51], (3) have 2 ADP (or low affinity) sites and 2 empty sites[48], and (4) are similarly highly processive when engaged with substrate[50]. These results therefore continue to illustrate the strength of PAN as a model system for the 19S to understand at least the fundamental functions of these machines. However, the 19S also has many associated non-ATPase subunits (unlike PAN) that form many contacts with the CC domains that almost certainly further influence heterogeneity of the CC conformations, (for example, the kink in the CC that has been observed in the Rpt1/2 CC of some recent 19S structures)[8,13,14].

The disulfide crosslinking methods we employ in this manuscript precisely show whether specific residues come within 3.4–4.6 Å of one another at the appropriate angle[47]. Therefore, any residues forming a disulfide crosslink can unambiguously be determined to have (at least momentarily) passed through a 1.2 Å space, whereas the available cryo-EM structures have a more limited resolution—especially in the CC domains. For example, as of the writing of this manuscript, the best cryo-EM average resolution of the 26S is 3.5 Å (5GJR), however, upon analyzing the structure's B-factors, we estimate that most portions of the Rpt CCs have resolutions of ~10–15Å[13]. Therefore, general trends in conformations these CC structures may adopt can be estimated by the available cryo-EM structures, but the disulfide crosslinking approach we have designed provides a much more precise tool for determining the proximity of specific residues, and thus CC

conformational states, in PAN. Using this disulfide crosslinking approach, we determined that PAN can adopt two different states: one with high activity (stabilized by crosslinking the C2 conformation) and one with lower activity (stabilized by crosslinking the C3 conformation) (Fig. 7d). Interestingly, these two states are mutually exclusive (i.e., they switch) from one another. This indicates that each CC must also have a secondary conformation that we could not capture via disulfide crosslinking here and so we can only speculate that these secondary states are perhaps unstructured.

The majority of ATPase ring complexes, including the proteasomal ATPases, are expected to hydrolyze ATP in a cyclical fashion. Based on this model, we expected that the CC domains would switch between their 3 different conformations during a full 360° ATP hydrolysis cycle. Indeed, the C2 and C3 conformations appear to fluctuate during ATP hydrolysis; however, crosslinking C1 of PAN (87C mutant) had absolutely no effect on ATPase activity or protein unfolding rates. This indicates that the C1 conformation can be static and unchanging during normal function. This was somewhat surprising since it is assumed that ATP is hydrolyzed in all subunits around the ring. How can we interpret sequential ordered ATP hydrolysis around the ring in light of this result? We envision two possibilities: (1) conformational changes in ATPase subunits that produce work due to ATP binding and hydrolysis are not directly and tightly correlated to the CC conformations or (2) ATP hydrolysis only occurs in four of the six ATPase subunits (i.e., those linked to the C2 and C3 conformations, which are more dynamic). We favor possibility #1 but cannot rule out #2. ATP hydrolysis in a subset of subunits in the 26S has been previously proposed elsewhere[52]. In case #1, how does regulation of ATPase activity by substrate binding fit in? The most stable state for any parallel CC is zipped and in-register. The fact that C2 and C3 are not in the C1 conformation indicates there are allosteric conformational restraints within the hexameric ring that only permit one C1 conformation, causing the others to adopt alternative states (i.e., C2 and C3). Thus C1 breaks PAN's hexameric symmetry and locks it into a conformationally asymmetric state. Interestingly, the C1 conformation relies on M87, and when it is mutated to an alanine, we observe an increase in PAN's basal ATPase activity (Fig. 6c). Based on this, we hypothesize that C1 may tend to pause cycles of ATP hydrolysis at this pair of subunits during normal PAN function. In this way, the C1 conformation may add a slight energy barrier to overcome before committing to another cycle of ATP hydrolysis, and the M87→A mutation reduces this energy barrier by slightly and locally destabilizing the proximal end of the C1 CC. An analogy would be a speed bump on a road, which is not intended to stop the flow of cars but to slow them. Similarly, substrate binding to the CC domains could lower this energy barrier and promote subsequent and more frequent rounds of ATP hydrolysis (i.e., accelerate ATP hydrolysis) for more rapid unfolding and degradation. It appears that PAN activation by M87A mutation or by substrate binding converge on a single activation mechanism, because both activate PAN, but when combined together their effect is not additive (activation by GFPssrA = activation by GFPssrA+activation by M87A; Fig. 6c). We postulate that this shared mechanism is via conformational alteration of PAN's CC domains, perhaps by shifting the population of PANs with C2 vs. C3 conformations. Prior analysis of the eukaryotic 26S proteasome via cryo-EM has shown that it can be found in low energy (resting) states and in activated (or substrate-bound like) states[1]. It is feasible that aspects of these CC conformational arrangements regulating ATPase activity could be conserved between archaea and eukaryotes. Further work will be required to solidify and differentiate between these models.

It is known that the 19S ATPases (like PAN) are activated by substrate binding to ubiquitin receptors, so we propose that CC domains in the 19S ATPases may adopt similar conformations to transmit allosteric signals to the AAA+ ATPase domains to increase ATP hydrolysis. Similarly, deubiquitinases associated with the CC domains (directly or indirectly) could allosterically regulate ATPase subunits to modulate timing of substrate engagement with deubiquitination[7,53,54]. Many PTMs have also been observed in the 26S, some that are activating and others that are inactivating[17–28]. Several of these PTMs have been observed in the N-terminal domains of the proteasomal ATPases. It is therefore likely that conformational changes in the CC domains regulated by these PTMs could also help to switch the proteasome between its activated and resting states. Thus, a fundamental mechanistic understanding of conformational changes in the proteasomal ATPase's CC domains will be essential to understanding the nuances of 26S regulation.

This study also overcomes a major limitation of PAN: its homomeric nature, which previously made it difficult to study its asymmetric properties like nucleotide binding. While others have circumvented this problem in distantly related homomeric ATP-dependent proteases (e.g., ClpX) by cleaving the N-domains to generate C-to-N-linked pseudohexamers to probe ring asymmetries, we determined that the N-domains of PAN have a crucial role in regulating the activity of the ATPase domains, so a parallel approach in PAN would be functionally deleterious. These crosslinkable PAN mutants now facilitate and open the door for future studies of structural asymmetries in PAN in native states, which serves as a starting point to further elucidate the mechanism of how proteasomal ATPases use the energy from ATP to drive protein degradation.

## Methods

**Materials, plasmids, and protein purification**. PAN, GFPssrA, and T20S (20S from *Thermoplasma acidophilum*) were prepared as in refs. [55,56]. Briefly, recombinant proteins were expressed in prsetA (PAN) or pet15b vectors (GFPssrA, T20S). BL21-DE3-RIL cells (Agilent) were used for PAN and T20S, and BL21 ClpX$^{-/-}$ cells (West Virginia University Protein Core) were used for GFPssrA. Cultures were induced with 0.5 mM isopropyl β-D-1-thiogalactopyranoside at $OD_{600} = 0.6$ for 4 h at 37 °C at 300 RPM. His-tagged GFPssrA and T20S were purified to homogeneity via Nickel-NTA Agarose (Thermofisher) in 50 mM Tris pH 7.5, 5% glycerol (v/v) and were eluted on a gradient of 10–300 mM imidazole over 10 column volumes. Purest fractions were pooled and further purified via size exclusion column (Superose 12 HR 10/30, GE) and tested for purity via SDS-PAGE and activity (gate opening for T20S, see below) or fluorescence (GFPssrA), and purest fractions were pooled. PAN variants were separated via two heat precipitation steps (60 °C for 30 min; 80 °C for 15 min) and spun at 15,000 RPM (SS34 Rotor, Sorvall), supernatants were pooled and further separated via anion exchange (HiTrap Q fast flow, GE, 5 ml) and size exclusion (Superose 12 HR 10/30, GE)), and purest fractions were determined via SDS-PAGE and/or activity (20S gate opening, unfolding, and/or ATPase assays, see below). For all purifications, saturating reducing reagent (1 mM dithiothreitol) was maintained at each purification step, then proteins were buffer exchanged (either by dialysis or PD-10 columns), snap frozen in liquid nitrogen, and stored at −80 °C until use. Expression vectors for the PAN CC mutants in pRSETA were generated by site-directed mutagenesis (QuikChange II Mutagenesis Kit) and were confirmed by sequencing (Sequetech). The purest available forms of ATP and ATPγS were purchased from Sigma and stored at −80 °C until use.

**Preparation of mini-spin G50 columns**. Twenty four hours prior to oxidation experiments, G50 (Illustra Sephadex G-50 Fine DNA Grade, GE) was preswollen in reaction buffer (50 mM Tris pH 7.5+5% v/v glycerol) at room temperature (1 g of G50 per 10 ml of reaction buffer). The day of a disulfide crosslinking experiment, 900 μl of G50 slurry was added to Pierce 0.8 ml Centrifuge Columns (Thermo Scientific). Mini-spin columns were equilibrated three times in fresh reaction buffer: 500 μl of reaction buffer was added to the column followed by centrifugation at $1500 \times g$ for 60 s, this was repeated two more times. On the last equilibration step, fresh buffer was added to column and the column was capped. Immediately prior to use, columns were uncapped and spun at $1500 \times g$ for 60 s. The size of the columns were measured (distance from the top of the tube to the center of the column), and any columns that were not within 1/2 mm of one another were discarded as was any residual G50 that had not properly packed into the column. The center of a typical column (with an initial 900 μl slurry) was 450–500 μl in

volume and was 6–7 mm below the top rim of the tube (1–2 mm below the threading marks).

**PAN CC mutant crosslinking and analysis**. PAN CC mutants were stored at −80 °C and were thawed on ice immediately prior to use. In every oxidation reaction, a reduced CC mutant control (1 mM DTT) and an oxidized WT control were run in tandem. All oxidation reactions were done in 50 mM Tris pH 7.5+5% glycerol (v/v) and were done in the absence of nucleotides, unless otherwise noted. The oxidizing reagent, TT, (sodium tetrathionate dehydrate ≥98%, Sigma) was used to oxidize the cysteines to allow for disulfide crosslink formation. Saturating TT concentration was determined for each CC mutant via dose–response, and this level of oxidizing reagent was used for crosslinking assays unless otherwise indicated. In general, 1 mM of TT for 1–2 h was sufficient to induce saturable crosslinking in the PAN CC mutants. Some PAN mutants have a tendency to form dodecamers (especially those carrying double cysteine mutations). Therefore, all PAN CC mutants were diluted to ≤0.25 mg/ml during disulfide crosslinking, which greatly minimized dodecamerization. Trace amounts of TT present in the samples upon addition to SDS sample buffer caused non-specific disulfide crosslinks to form since SDS denatures PAN. Therefore, after oxidation but prior to SDS-PAGE analysis each PAN sample was carefully desalted on G50 mini-spin columns. Twenty microliters of each sample was added to mini-spin G50 columns, which were prepared as described above. Following desalting, the concentrations of each sample was recalculated via Bradford assay immediately after desalting. Approximately, 50% of PAN was recovered after each desalting column to assure that TT was completely removed. Crosslinking of every sample was confirmed via SDS-PAGE analysis prior to use. Three micrograms per well was used for SDS-PAGE analysis. Following SDS-PAGE, gels were rinsed 3 times with double deionized water, stained with Bio-Safe Coomassie G-250 Stain (Biorad) for 1 h, rinsed in 100 ml of double deionized water for 1 h, and then 20 ml of 20% NaCl (w/v) was added for at least 1 h. Gels were then imaged on a Syngene GBox Imager. We found using this strict staining/rinsing method that the densitometry signal remained linear between 0.15 and 5 μg PAN (Supplementary Fig. 8), which allowed for accurate calculation of the relative amounts of PAN within this range. Disulfide crosslink ratio was determined by calculating the pixel intensity of monomers vs. dimers in ImageJ after subtracting control densities. Data are means of at least 3 independent experiments ($n \geq 3$) ± standard deviations. Full-length lanes of all crosslinking experiments in PAN mutants from which major conclusions are drawn (WT-PAN, PAN-87C, PAN-59C, PAN-87+73C, and PAN-87+73+59C) are provided in Supplementary Fig. 1.

**Partial proteolysis of PAN**. In all, 0.4 mg/ml of PAN was mixed with increasing amounts of Trypsin (0–2 μg per 40 μl rxn), incubated for 1 h at room temperature in reaction buffer (50 mM Tris pH 7.5, 5% v/v glycerol), and reactions were quenched with the manufacturer's recommendation amounts (1:100) of Halt Protease Inhibitor Cocktail (Thermo Scientific). Samples were checked for CC-OB domain fragment via Native-PAGE, and samples containing CC-OB domain fragment were pooled and injected onto a size exclusion column (Superose 12 HR 10/30, GE). Immediately after elution, 1:100 Halt Protease Inhibitor Cocktail (Thermo Scientific) was added to each fraction. Peaks were analyzed via SDS-PAGE and Native-PAGE and Fraction 1 (containing near-full-length PAN) and Fraction 3 (containing CC-OB fragment) were pooled. Fractions 1 and 3 were incubated with either 1 mM DTT (reduced sample) or 1 mM TT (oxidized sample) for 1 h at room temperature. After desalting (as described above), samples were run on SDS-PAGE and visualized via silver stain (Pierce Silver Stain Kit, Thermo Scientific). Representative images from four independent experiments are shown.

**PAN-M87C crosslinking timecourse**. In all, 1 mg/ml PAN-M87C was incubated overnight at 4 °C with reducing agent (DTT). Reactions were carried out at −17 °C, which was achieved using 11% NaCl in ice (w/w). This temperature was maintained for 6 h, after which the temperature had risen to −15 °C. Samples contained 50% glycerol to prevent freezing[57]. Immediately prior to an experiment, samples were desalted in G50 (see "PAN CC mutant crosslinking and analysis" section), and diluted to 0.25 mg/ml in 50 mM Tris pH 7.5+50% glycerol. In all, 1 mM TT was added to start the oxidation reaction, and at each time point a sample was immediately desalted and added to a 1× SDS sample buffer for analysis via SDS-PAGE.

**ATPase, GFP unfolding, and 20S gate-opening assays**. All experiments were performed at 37 °C with absorbance or fluorescence measured in a Biotek 96-well plate reader. Data are means of at least three independent experiments ($n \geq 3$) ± standard deviations. ATP hydrolysis was measured by using an NADH-coupled ATP regenerating system (50 mM Tris pH 7.5, 5% glycerol (v/v), 20 mM MgCl$_2$, 2 U/μl pyruvate kinase, 2 U/μl lactate dehydrogenase, 3 mM phosphoenolpyruvate, and 0.2 mg/ml NADH) by reading the loss of NADH absorbance at 340 nm every 20 s. In all, 2 mM ATP was used unless otherwise indicated (e.g., ATP dose–response curves).

For the unfolding experiments, reaction buffer (50 mM Tris pH 7.5, 5% glycerol (v/v)) was incubated with 20 mM MgCl$_2$, 50 nM PAN, 400 nM T20S and 0.2 nM GFPssrA and 2 mM ATP. Green fluorescent protein (GFP) fluorescence loss (ex/

em: 485/510) was measured every 20 s in a Biotek 96-well plate reader at 37 °C to obtain unfolding rates. Data are means of at least three independent experiments ($n \geq 3$) ± standard deviations.

Gate opening was measured in the reaction buffer with the archaeal T20S (3 nM), PAN mutants (200 nM), and $MgCl_2$ (20 mM) using the internally quenched fluorogenic peptide substrate (LFP)[56]. LFP was dissolved in dimethylsulfoxide and used at a final concentration of 10 μM in the presence or absence of 10 μM ATPγS. LFP contains a fluorescent reporter (MCA) at the N-terminus and a quenching group (DNP) at the C-terminus. Upon cleavage of the peptide by the 20S proteasome, MCA is released and an increase in fluorescence can be observed at ex/ em: 325/393) can be observed. Rate of fluorescence increase (ex: 325 em: 393) was measured every 20 s in a Biotek 96-well plate reader to determine the rate of 20S activation (gate opening) by PAN.

**19S and PAN structure analysis**. 19S cryo-EM structures (e.g., PDB: 4CR4 [https://doi.org/10.2210/pdb4CR4/pdb]) and PAN CC-OB crystal structures (PDB: 3H43 [https://doi.org/10.2210/pdb3H43/pdb]) were analyzed in Pymol. β-Carbon distances were measured using the Measurement tool.

**PAN MS analysis**. Uncrosslinked and crosslinked PAN-M87C samples were trypsinized and sent for analysis by the Scripps Center for Metabolomics in La Jolla, CA. Crosslinked samples were analyzed in a non-reducing environment. Peptides were analyzed by reverse-phase chromatography prior to MS analysis using the following method. Nanoelectrospray capillary column tips were made in-house by using a P-100 laser puller (Sutter Instruments). The columns were packed with Zorbax SB-C18 stationary phase (Agilent) purchased in bulk (5-mm particles, with a 15-cm length and a 75-mm inner diameter). The reverse-phase gradient separation was performed by using water and acetonitrile (0.1% formic acid) as the mobile phases. The gradient consisted of 5% acetonitrile for 10 min followed by a gradient to 8% acetonitrile for 5 min, 35% acetonitrile for 113 min, 55% acetonitrile for 12 min, and 95% acetonitrile for 15 min and re-equilibrated with 5% acetonitrile for 15 min.

Data-dependent MS/MS data were obtained with an LTQ linear ion trap mass spectrometer using a home-built nanoelectrospray source at 2 kV at the tip. One MS spectrum was followed by four MS/MS scans on the most abundant ions after the application of the dynamic exclusion list. Tandem mass spectra were extracted by use of the Xcalibur software. All MS/MS samples were analyzed by using Mascot (version 2.1.04; Matrix Science, London, United Kingdom) with *Staphylococcus epidermidis* proteins contained in the NCBInr protein database, assuming the digestion enzyme trypsin. Mascot was searched with a fragment ion mass tolerance of 0.80 Da and a parent ion tolerance of 2.0 Da; identification was done at the 95% confidence level with a calculated false-positive rate of <1% as determined by using a reversed concatenated protein database. Peptide identifications were accepted if they could be established at a >95.0% probability as specified by the Peptide Prophet algorithm. Protein identifications were accepted if they could be established at a >99.0% probability and contained at least two identified peptides as specified by the Protein Prophet algorithm. Proteins that contained similar peptides and could not be differentiated based on MS/MS analysis alone were grouped to satisfy the principles of parsimony.

**Statistical analysis**. The data were analyzed using an unpaired Student's *T*-test (Sigmaplot). For all statistical analyses, a value of $P < 0.05$ was considered significant. Minimum sample size $n = 3$ was chosen to ensure reproducibility and to generate standard deviations.

**Data availability**. The data supporting the findings of this study are available from the corresponding author upon request.

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

## Acknowledgements

We thank the members of the Smith laboratory for helpful and valuable discussions and the protein core at WVU for their services. This work was supported by NIH-R01GM107129 to D.M.S. and by F31GM115171 to A.S.

## Author contributions

A.S. designed all experiments and analyzed data (with input from D.M.S.). A.S. and E.J.B. purified all proteins and performed all experiments in the manuscript. Manuscript preparation was done by A.S. and D.M.S. All the authors reviewed the results and approved the final version of this manuscript.

## Additional information

**Competing interests:** The authors declare no competing interests.

