## [Peer Review File · Nature Communications]

Reviewers' comments:

Reviewer #1 (Remarks to the Author):

ATP-dependent protein degradation is executed by the 26S proteasome, which includes a hetero-hexameric complex (Rpt1-6) of AAA+ proteins that unfold and thread substrates into the associated proteolytic 20S chamber. PAN represent the archaeal homologue of Rpt subunits but forms a homo-hexamer and therefore offers a simplified system to study AAA+ protein function. Rpt1-6 as well as PAN subunits include N-terminal coiled-coil domains (CCDs), which have been shown before to be crucial for activity. Here, the authors analyzed the conformational states of the CCDs by site-specific disulfide crosslinking. They show that CCDs adopt different structures, involving three different types (in-register zipped, partially unzipped and 2-heptad slide zipped) of coiled-coil formation in an asymmetric PAN hexamer. The different PAN CCD structures nicely correspond to the CCD structures of Rpt1-6 determined before by cryoEM. One CCD conformation (C1) is suggested to be "fixed", whereas C2 and C3 conformations are proposed to fluctuate, ultimately defining "resting" and "activated" PAN states with low and high ATPase activities. The evidence for such conformational switch is indirect.

The disulfide crosslinking experiments are well executed and the experimental evidence for three different CCD structural states is convincing. Linking these CCD structural states to specific PAN activity states is, however, problematic. It remains unclear how CCD conformations might control ATPase function. Furthermore, AAA+ proteins including PAN are suggested to represent cycling machines involving e.g. a sequential mode of ATP hydrolysis and substrate threading (as also suggested by the authors: Kim et al., Nature Comm 2015). In case of PAN this involves continuous switching of individual AAA+ subunits from an empty state to states with either high or low nucleotide affinities. How do the authors envision ATPase control by CCDs involving a fixed C1 conformation if associated AAA domains constantly change their activity states? These crucial mechanistic aspects are not covered by the presented work but seem important for acceptance. The authors are therefore asked to provide further data pointing at the mechanism of PAN activity control by CCDs.

Further points:

- 1) In case of Rpt1-6 the different CCDs also interact with components of the 19S cap. Are the different structural states of these CCDs directly involved in partner binding? Do the authors envision that these partners impact Rpt1-6 ATPase activities by stabilizing CCD conformations?
- 2) How do specific nucleotide states (empty, all ADP, all ATP (using ATP γ S).) affect the crosslinking pattern (CCD conformations)? In view of PAN subunits cycling between different nucleotide states such experiment might be informative. Along the same line, does substrate binding and interaction with the peptidase influence CCD crosslinking?

- 3) Fig. 2E: How much SDS was added in the crosslinking experiments? Does SDS addition affect hexamer formation and ATPase activity?
- 4) Fig. 3B/C: the authors refer to the proteolytic fragment including the CCD as monomer, but the fragment seems to constitutively form a dimer in SDS-PAGE. Furthermore, the authors should reduce the crosslinked (dimeric) fragment (Fig. 3C) to confirm specificity.
- 5) Page 22: the end of the results section is missing

Reviewer #2 (Remarks to the Author):

This is an interesting report on the conformational variability in the N-terminal domains of the archaeal AAA protein, PAN, a homolog of proteasomal ATPases from higher organisms. Using site directed mutagenesis to engineer proteins with strategically placed cysteine residues, the authors examined direct disulfide cross-linking patterns to show that the N-terminal coiled coil (CC) domains of PAN, which are needed to engage and unfold protein substrates, exist in different conformational states. They propose that the different conformational states are required for efficient unfolding of substrates and that the different conformations reflect an underlying asymmetry in PAN, and by extension other AAA proteins with similar functions, that is necessary for their activity. AAA proteins are ringed complexes that can be composed of identical or similar proteins. Nevertheless, studies of AAA proteins with diverse functions and from diverse organisms have given rise to an understanding that in executing their mechanical functions the subunits within the rings adopt different conformational states at any given time and that they cycle or otherwise transition between these states while hydrolyzing ATP and variously binding and letting go of polypeptide chains within the substrate protein. The current findings, if completely validated, would constitute a significant advance in helping to substantiate the current models of the mechanism and extend the notion of asymmetry to the linked N-terminal coiled-coil domain of PAN and similar enzymes.

The experiments performed in this study were carefully planned and elegantly executed. By placing cysteine residues at specific locations within the heptad repeats of the CC domain and quantifying the degree of spontaneous cross-linking, the authors showed that each of the three CC domains made up of helices from adjacent subunits have different contacts. Using spontaneous disulfide formation as a measure of closeness, the authors show that one CC has all five heptads in register, one CC has only its distal heptads in register, and that the third CC has the heptads shifted so that heptads 1,2, and 3 and in register with heptad 3,4 and 5 in its partner. While the logic of the experiments and their interpretation is unassailable, it remains that there is no direct chemical demonstration of disulfide bond formation and no chemical identification of the polypeptide chains

in which the cross-linked residues reside. The conclusions of this study are too important to be based solely on correspondence between the calculated probability of forming a disulfide link and the apparent percentage of cross-linking observed. The cross-linked proteins need to be isolated, digested, and methods employed to identify chemically the residues forming the cross-link. Until that is done other explanations cannot be ruled out with complete confidence.

The authors should also address the question of the degree to which allosteric effects upon cross-linking of the first CC in a hexamer might affect the ability to cross-linking other CCs. The presence of an allosteric influence would not invalidate the model and might even enrich it and it should be looked into.

Reviewer #3 (Remarks to the Author):

Overall, manuscript is clearly written and well-focused on the important topic of the fundamental aspects of proteasome function. The findings advance the field by providing an insight into the conformational states of the coiled coil domains of the archaeal PAN, a homolog of the Rpt1-6 AAA ATPases of 26S proteasomes.

Comments regarding experimentation:

P7 last two lines state [both oxidized and reduced forms of WT and M87C PAN formed hexamers via native-PAGE (Fig. S1A)] however, only the oxidized form of the PANs are displayed in Fig. S1A.

Fig. S1A: The data, used to argue the WT and variant PANs are all hexamers, are not convincing. The resolution of gel electrophoresis cannot be assessed, as only one protein band per lane is detected. Some type of control protein (e.g., PAN subcomplexes) is needed in the gel. If the PANs were similarly purified on a high-resolution gel filtration column calibrated with molecular mass standards, this type of information would be more convincing than the gels.

P9-10: disagree with statement that the evidence (provided at this point in the manuscript P9 In 10-11 which is a dose response with tetrathionate) indicates [that all cysteines within proximity to one another had successfully crosslinked, thus excluding possibility #1]. A control protein is needed that harbors a cysteine residue in similarly close-proximity to those of the PAN CC that can be fully cross-linked and fully reduced under the experimental conditions of this study to rule out possibility #1.

P10 Fig. 2E: the finding that in the presence of denaturant up to 96% of PAN proteins are dimers (vs. monomeric) is encouraging that the cross-linking conditions can capture any dimers that may be present. However, the overall level of PAN protein detected in the gel seems to have decreased upon increasing denaturant (e.g., the last lane on the right seems to have less protein than the first lane on the left in terms of total dimer + monomer). Please clarify. Is there any PAN protein retained in the well? The PAN could be x-linked into higher order structures that may be at low levels throughout the gel and, thus, contribute to the reduction in monomer levels. Are native cysteine residues found in the PAN? MJ1176 has 3 cysteines based on UniProtKB Q58576. As the PAN is denatured, how did you rule out that the native cysteine residues do not account for the reduced level of PAN monomers detected under oxidizing conditions?

P10 In 18-20: [while subcomplex II (F2) contained a 30-35 kDa fragment..] while unimportant to the conclusions of this manuscript - I cannot find these results which are described in the text.

P10-11: the experiment associated with Fig. 3 is inconclusive in the absence of a PAN WT control. Panel C is particularly troubling. Based on the current evidence, an alternative explanation could be that the isolated CC-OB domain is tightly associated and cannot be fully denatured under SDS-PAGE gel conditions. The purity of the F3 subcomplex I of PAN M87C is also problematic. The proteolytic fragments observed in the SDS-PAGE gel in panel C, while indicated to be limited to high molecular mass fragments, are found to differ in profile between the oxidized and reduced states of the PAN M87C CC-OB domain and could also account for band indicated as a 'monomer' only based on migration in the gel. Thus, the oxidation state could have nothing to do with the migration of the M87C CC-OB domain fragment by SDS-PAGE. The claim that 100% of the CC-OB domain was cross-linked under oxidizing conditions (vs. the 33% observed for the full length) is hard to conclude – since the reduced state of the CC-OB is at 75% dimer vs. the full length at 0% dimer as observed by SDS-PAGE.

Statement regarding experimental reproducibility needs to be included for Fig. 3, Fig. S1-3 (please provide the number of experimental, biological, and technical replicates in the legend).

Please indicate p-values on each bar graph by * or some other standard notation (e.g., Fig. 4B states p-values for all points related to 87C were <0.001, yet the % crosslinking of M87C, M80C and M73C all appear to be around 33% and are argued so in the text. p-values are particularly needed for Fig. 6C since the claim is that PAN WT and M87A differ in the rate of ATP hydrolysis in the absence of substrate.

P16: the authors discuss register shifts in terms of one or more 'heptad' (defined as a group or set of seven). This type of nomenclature is confusing to me – as a shift of one or more amino acid residue within the heptad is indicated by the data and within the figures. For example, a register shift of one heptad really seems to mean a register shift of one amino acid within the heptad.

P20 In 8: disagree with term [must] - the data instead support the term [can]

P20 In 16: the authors do not provide data to demonstrate the substrate bound state of PAN.

P22 In 6-9: one scientific hurdle is that the 87C is common to locking both C1 (87C-87C) and C3 (87C-73C) states which makes the data less direct

Regarding locking PAN in different functional states - more details need to be provided in the methods section regarding the cross-linking conditions in terms of presence vs. absence of ATP or other nucleotide analog, temperature, buffer conditions, and range of protein concentration (only know less than or equal to 0.25 mg/ml). This information will help guide the reader regarding the functional state of PAN during the crosslinking.

P24 In 6: disagree with statement [the present student provides the first direct structural evidence of this..] - the evidence is supportive but not direct or structural. In fact the statement P24 In 12-13 regarding a previous cryoEM study seems to provide direct structural evidence for asymmetry of CC domains of the AAA ATPases of the 26S proteasome.

Figures:

Fig. 2. Need to clarify in Fig. 2 legend that Fig. S5 supports the quantitative analysis of PAN protein levels in the Fig. 2 gels and subsequent gels.

Fig. 2B,2C-E, 3B-C, 4B, 5B-C, 7A, S2A, S2C and S5 are missing the molecular mass standards - yet the reader is expected to analyze Fig. 3C in terms of Mr and compare these molecular masses to the other gels that display the x-linked dimer and monomer. Providing only arrows that define each protein band in terms of dimer, monomer, etc. leads to assumptions that may not be justified. Indicating the migration of the Mr standards for the SDS-PAGE gels is important.

Fig. 3B, Gel filtration molecular weights are discussed in text but only presented in terms of elution volume in the figure. Furthermore, the methods section P31 In12 includes no definition of the protein standards used for calibration and no listing of the dimensions of the Superose 12 column.

Fig. 4 presented out-of-order (panel ACB) in the text.

Fig. 6A-C, 7B-C, Fig. S3, Table S1-2: what do the numbers represent in terms of units of activity and Vmax? e.g., in panel 6A the ATPase rate is listed as 60 ATP per PAN per min? is this 60 moles of ATP per mole of PAN per min? Please define 100% in the footnote of Table S2.

Fig. S1: define the rate of LFP hydrolysis at 1-fold in terms of units/mg in the figure legend.

Minor editing issues:

Some figures are out of order in terms of presentation and discussion within the manuscript (see specific comments)

P1: briefly clarify early in the manuscript that each of the three-identical coiled coil (CC) domains discussed in the abstract are formed by the N-termini of two separate PANs (otherwise the reader is left wondering until P5 In 11-12)

P1 In 13: define CC within abstract and text

P1 In 12: [26S ATPases] need to clarify that the ATPases are not 26S

P2 In 10: the proteasome? which type 26S?

P2 In 14+: [19S ATPase] need to clarify that the ATPases are not 19S

P2 In 14+: define what is meant by Rpt6/3 CC, Rpt1/2 CC and Rpt4/5 CC? see comment regarding the need to clarify that CC domains are each formed by the N-termini of two separate Rpt subunits earlier in the manuscript

P2 In 19: define DUB and clarify which type of proteasome is meant by [the proteasome]

P2 In 23-24: [This alone indicates that CC domains play fundamental roles in proteasome function] what evidence is this alone referring to? Am unclear which study alone is providing such striking evidence.

P2 last line: please define [their] in [their functional importance]

P3 In 1: switch between 26S vs. 26S proteasome is confusing

P3 In 16-21: need to cite reference(s) to support the important points made in the first three sentences of this second paragraph

P4 In 17: may not want to state so strongly - as the model substrates used in this study to demo PAN: proteasome function (GFPssrA and LFP-Amc) are not tagged with ubiquitin-like proteins or found in the archaeal cell

P5 In 15-16 and 18: spell out species names upon first use and italicize throughout

Fig. S1A and other figures: please define 87C, 80C etc. in the figure legends (particularly the first figure legend). For example, that 87C refers to M87C, etc. Otherwise, the designation is easily confused with temperature in degrees Celsius.

Fig. S1A and text: need to define LFP (indicating that it is AMC linked)

P9 reference to Fig. 6A-B is out of order and confusing

P9 In 4-5: recommend modifying to state: PAN could exist in at least two different conformational populations including one that is 'crosslinkable' and another that is not...

P18 20-21: [expected] used twice within one sentence

P21 In 3: please reword the sentence – since the rationale that [crosslinking the C1 conformation has no effect on activity]- does not support the point that [stabilizing the C3 conformation by crosslinking stabilizes a functional, but low activity state of PAN].

P28 In 21: T20S define

P29 In 8+: define % in terms of v/v or w/v

Throughout the manuscript - be sure to include space between the number and units - e.g, 100 ml vs. 100ml and avoid random capital letters (e.g., Pyruvate Kinase and Lactate dehydrogenase mid-sentence).

P31 - Methods for partial proteolysis of PAN are not well defined. PAN was incubated with trypsin for 1 h in what buffer? How much of the quenching solution was added? What was the type of

Superose 12 column (10/300 GL? this info will provide column dimensions), what type of buffer was the column equilibrated in?

References need a significant amount of formatting: ref. 4, 5,9,11,12,15,21,23,28,30,38,41,43,45,47,49 all need some type of correction e.g, use title case lettering, include volume numbers, italicize species names where appropriate, proofread for font errors such as ??, only include the article title once.

Reviewers' comments:

Reviewer #1 (Remarks to the Author):

ATP-dependent protein degradation is executed by the 26S proteasome, which includes a hetero-hexameric complex (Rpt1-6) of AAA+ proteins that unfold and thread substrates into the associated proteolytic 20S chamber. PAN represent the archaeal homologue of Rpt subunits but forms a homo-hexamer and therefore offers a simplified system to study AAA+ protein function. Rpt1-6 as well as PAN subunits include N-terminal coiled-coil domains (CCDs), which have been shown before to be crucial for activity. Here, the authors analyzed the conformational states of the CCDs by site-specific disulfide crosslinking. They show that CCDs adopt different structures, involving three different types (in-register zipped, partially unzipped and 2-heptad slide zipped) of coiled-coil formation in an asymmetric PAN hexamer. The different PAN CCD structures nicely correspond to the CCD structures of Rpt1-6 determined before by cryoEM. One CCD conformation (C1) is suggested to be “fixed”, whereas C2 and C3 conformations are proposed to fluctuate, ultimately defining “resting” and “activated” PAN states with low and high ATPase activities. The evidence for such conformational switch is indirect.

The structural and functional evidence we provide for a conformational switching in PAN's CC domains is fairly direct

- 1) Cysteines can only crosslink when in proximity, and when C2 is crosslinked we cannot crosslink C3, however C3 can be crosslinked independently of C2. This alone is direct evidence of 2 distinct conformational states at the engineered cysteine positions.
- 2) The C3 crosslinked PAN is functionally distinct from the C2 crosslinked PAN.

The disulfide crosslinking experiments are well executed and the experimental evidence for three different CCD structural states is convincing. Linking these CCD structural states to specific PAN activity states is, however, problematic. It remains unclear how CCD conformations might control ATPase function.

We agree that it is unclear exactly how these different CC conformations are able to regulate ATP hydrolysis. Indeed, we are very interested in how exactly these conformations are allosterically connected to the AAA+ ATPase domains. The purpose of this study was to identify that the CC domains indeed adopt different conformations, and to show that these conformations regulate ATPase function. Mechanistic determination of how the CC domains regulate function of this complex machine will not be trivial and will require its own independent study, and thus is outside of the scope of this already lengthy study.

Furthermore, AAA+ proteins including PAN are suggested to represent cycling machines involving e.g. a sequential mode of ATP hydrolysis and substrate threading (as also suggested by the authors: Kim et al., Nature Comm 2015). In case of PAN this involves continuous switching of individual AAA+ subunits from an empty state to states with either high or low

nucleotide affinities. How do the authors envision ATPase control by CCDs involving a fixed C1 conformation if associated AAA domains constantly change their activity states?

It was quite surprising to us that some of the CCs can remain fixed and yet have normal ATP hydrolysis and substrate unfolding. Based on our previous cyclical hydrolysis model (Kim et al. *Nat. Comm.* 2015), we initially expected that the CC domains would switch between their 3 different conformations during a full hydrolysis cycle of all 6 subunits. However, the C1 conformation seems to remain static and unchanging during normal function. See our response to the next point for further discussion of how we envision the CC domains can control ATPase domains.

These crucial mechanistic aspects are not covered by the presented work but seem important for acceptance. The authors are therefore asked to provide further data pointing at the mechanism of PAN activity control by CCDs.

We also are very interested in how the CC domains are able to control PAN activity. However, such mechanistic details of how ATPase rates are regulated in basal states have not been worked out for any AAA ATPase. Such fundamentals will need to be clarified before we can expect to understand mechanism behind the throttling of activity. Such efforts will require further independent study. However to try to address this point, in the text, we do provide 2 models that would explain how ATPase rates are controlled by a fixed C1 conformation. We have edited the text in an attempt to clarify that we are proposing 2 possible models that shed light on the potential mechanisms of regulation without overreaching. Quoted here:

“The majority of ATPase ring complexes, including the proteasomal ATPases, are expected to hydrolyze ATP in a cyclical fashion. Based on this model we expected that the CC domains would switch between their 3 different conformations during a full 360° ATP hydrolysis cycle. Indeed, the C2 and C3 conformations appear to fluctuate during ATP hydrolysis; however, crosslinking the C1 conformation of PAN (87C mutant) had absolutely no effect on ATPase activity or protein unfolding rates. This indicates that the C1 conformation can be static and unchanging during normal function. This was somewhat of a surprise since it’s assumed that ATP is hydrolyzed in all subunits around the ring. How can we interpret sequential ordered ATP hydrolase around the ring in light of this result? We envision 2 possibilities: 1) conformational changes in the ATPase subunits that produce work due to ATP-binding and hydrolysis are not directly and tightly correlated to the CC conformations, or 2) ATP hydrolysis only occurs in four of the six ATPase subunits (i.e. those linked to the C2 and C3 conformations, which are more dynamic). We favor possibility #1 but cannot rule out #2. ATP hydrolysis in a subset of subunits in the 26S has been previously proposed elsewhere (Beckwith et al. 2013). In case #1, how does regulation of ATPase activity by substrate binding fit in? The most stable state for any parallel CC domain is zipped and in register. The fact that C2 and C3 are not in the C1 conformation indicates there are allosteric conformational restraints within the hexameric ring that only permit one C1 conformation, causing the others to adopt alternative states (i.e. C2 and C3). Thus, the C1 conformation breaks PAN’s hexameric symmetry and “locks” it into a conformationally asymmetric state. Interestingly, the C1 conformation relies on M87, and when

it is mutated to an alanine we observe an increase in PAN's basal ATPase activity (Fig 6C). Based on this we hypothesize that the C1 conformation may tend to pause cycles of ATP hydrolysis at this pair of subunits during normal PAN function. In this way the C1 conformation may add a slight energy barrier to overcome before committing to another cycle of ATP hydrolysis, and the M87→A mutation reduces this energy barrier by slightly and locally destabilizing the proximal end of the C1 CC. An analogy would be a speed bump on a road, which is not intended to stop the flow of cars, but to slow them. Similarly, substrate binding to the CC domains could lower this "energy barrier" and promote subsequent and more frequent rounds of ATP-hydrolysis (i.e. accelerate ATP-hydrolysis) for more rapid unfolding and degradation. It appears that PAN activation by M87A mutation or by substrate binding converge on a single "activation" mechanism, because both activate PAN, but when combined together their effect is not additive (activation by GFP-ssrA = activation by GFP-ssrA + activation by M87A; Fig 6C). We postulate that this shared mechanism is via conformational alteration of PAN's CC domains, perhaps by shifting the population of PANs with C2 versus C3 conformations. Prior analysis of the eukaryotic 26S proteasome via cryo-EM, has shown that the 26S proteasome can be found in low energy or "resting" states and in "activated" (or substrate-bound like) states (Matyskiela et al. 2013). It's feasible that aspects of these CC conformational arrangements regulating ATPase activity could be conserved between archaea and eukaryotes. Further work will be required to solidify and differentiate between these models."

Further points:

1) In case of Rpt1-6 the different CCDs also interact with components of the 19S cap. Are the different structural states of these CCDs directly involved in partner binding? Do the authors envision that these partners impact Rpt1-6 ATPase activities by stabilizing CCD conformations?

We cannot definitively say whether different structural states of the Rpt1-6 CCs are involved in partner binding, since this manuscript experimentally focused solely on PAN. However, we do expect that different factors (either Rpn subunit interactions, substrate binding, or post-translational modifications) could influence CC conformations, and thus ATPase activity in the 19S ATPases. To address this idea, in the last sentence of the 1st discussion paragraph we state:

"the 19S also has many associated non-ATPase subunits (unlike PAN) that form many contacts with the CC domains that almost certainly further influence heterogeneity of the CC conformations".

We believe that this sentence left open the possibility of Rpn subunits influencing CC conformations without overstepping the main conclusion we made from this manuscript, which only looked at the CCs in PAN.

2) How do specific nucleotide states (empty, all ADP, all ATP (using ATPγS).) affect the crosslinking pattern (CCD conformations)? In view of PAN subunits cycling between different nucleotide states such experiment might be informative.

We agree and indeed we performed many related experiments with PAN in different nucleotide states. We found that addition of nucleotides did not affect crosslinking levels that we observed. See **Fig R1.1 and R1.2** examples of M87C crosslinking in the Apo, ADP, ATP, or ATP_γS state. **Fig R1.1** are endpoint experiments carried out at room temperature, and no difference was observed except for a modest decrease in crosslinking in the high ATP_γS state [which has previously been shown to force PAN into an “unnatural” 4-nucleotide bound state with suboptimal activity (Smith et al *Cell* 2011)] (**Fig R1.1**). We thank the reviewer for raising this concern, since upon reanalysis of these data we believe that this will be of great interest to the readers, and have added a supplemental figure (**Fig S3, which contains Fig R1.1 and R1.2 from this document**) accompanied with a short discussion in the text.

Figure R1.1:

Interestingly, when the disulfide crosslinking reaction is slowed down by carrying out timepoint experiments at -17°C, we do observe a small effect on the rate of crosslinking in PAN-M87C in the presence of ADP (**Fig R1.2**). This effect is not resolvable in the endpoint experiments of **Fig**

Figure R1.2:

R1.1 since the reaction goes to completion so quickly. Thus, while saturating ADP levels do appear to slightly disorganize the C1 conformation (i.e. slows the rate of M87C crosslinking), this effect is very modest and was only observed very low temperatures, and binding of ADP did not completely restrict PAN from entering the C1 conformation, perhaps due to the fast on/off rate of ADP (seconds) relative to the timecourse of this experiment (minutes to hours).

Along the same line, does substrate binding and interaction with the peptidase influence CCD crosslinking?

In short, substrate binding does not affect the extent of M87C crosslinking (C1 conformation; see **Figure R1.3** below-left panel). Also, binding of the peptidase (PAN-20S complex) during GFP-ssrA degradation does not affect the extent of M87C crosslinking (M87C compared to WT; **Figure R1.3** right panel). The details of these experiments are presented below. It is, however, possible (as discussed) that substrate binding will affect the conformational changes of the C2 and/or C3 CC domains in more nuanced ways, but we do not have evidence of this.

Figure R1.3:

Experimental details for Figure R1.3.

Left panel: 100 nM PAN (WT or M87C mutants) was preincubated with 3 μ M GFPssrA for 1 hour in the absence of protease inhibitors (Left panel; 2 μ M ATP γ S and 10mM MgCl₂ were also present in the assay). Then, saturating oxidizing agent (Tetrathionate, 1mM) was added and incubated for 1 hour at room temperature then overnight at 4 $^{\circ}$, desalted, and analyzed via SDS-PAGE. 30% dimers were observed in the M87C samples, indicating that preincubation of substrate did not affect M87C disulfide crosslinking.

Right Panel: GFP-ssrA unfolding was analyzed with a fluorogenic unfoldase assay and degradation was followed via SDS-PAGE. Oxidized PAN was preincubated with T20S and GFPssrA for 20 minutes in a non-reducing environment at 37 $^{\circ}$ (this assay also contained all ATP regeneration assay components except for ATP itself). After a 20 minute incubation, 2mM ATP was added to start the reaction. After GFP fluorescence was monitored for 20min the fluorescently analyzed samples were desalted and analyzed via SDS-PAGE. Note the fluorescence of GFP in the 3 samples in the inset, and compare to the GFP band in the gel. GFPssrA is present in the “No PAN” lane (no degradation), but the presence of WT PAN or PAN-M87C no GFP band is visible, which demonstrates degradation by the PAN-20S complex. Upon analysis of PAN-M87C—it forms \sim 1/3 dimers, similar to conditions without the 20S present.

Taken together, we conclude that preincubation with GFPssrA does not affect M87C crosslinking nor does binding of the 20S to PAN. Further, the WT PAN-20S complex and oxidized M87C PAN-20S complex unfold and degrade GFP-ssrA to similar extents (M87C was slightly faster in this experiment).

3) Fig. 2E: How much SDS was added in the crosslinking experiments?

The SDS amounts on gel are as follows: 0.62%, 0.31%, 0.062%, 0.031%, 0.0062%, 0.0031%, 0.00062%, 0.000062%. We have now included this information in the figure legend.

Does SDS addition affect hexamer formation and ATPase activity?

Indeed it did, as expected. We tested hexamer formation via native gels alongside SDS-PAGE in the SDS denaturant assay (**Fig R1.4**). As illustrated below, as increasing amounts of SDS are added (**Fig R1.4, top panel**), quaternary structure is perturbed, as expected (**Fig R1.4, bottom panel**). Note that formation of \sim 95% dimers in the SDS gel (e.g. the point at which almost all of PAN's CCs crosslink) corresponds to the perturbation of PAN's quaternary structure observed in the native gel, as expected. Since only one C1 conformation can be found in PAN under native conditions, it is expected that structural denaturation would be required to allow all three CC domains to crosslink in the C1 conformation, which was the purpose of the experiment.

Figure R1.4:

Figure R1.4: Increasing amounts of SDS added to 0.25mg/ml PAN-M87C following incubation with 1mM tetrathionate for 1 hour. Top panel is SDS-PAGE (same as **Figure 2E** in main manuscript) and bottom panel is Native PAGE of these same samples.

4) Fig. 3B/C: the authors refer to the proteolytic fragment including the CCD as monomer, but the fragment seems to constitutively form a dimer in SDS-PAGE. Furthermore, the authors should re-reduce the crosslinked (dimeric) fragment (Fig. 3C) to confirm specificity.

As requested, we have changed the text to clarify that bands corresponding to the molecular weight of CC-OB monomers and dimers were found:

“We analyzed both subcomplexes via SDS-PAGE and found that Subcomplex I (F3) contained an ~8 kDa fragment (the expected size of a single CC-OB domain), and a ~16 kDa fragment (the expected size of a CC-OB dimer) (Fig. 3C)”.

In addition, we have added the molecular weight markers to Figure 3C for clarity of specificity. See also our response to Reviewer #3 (“P10-11” and “3B”) for a discussion of similar points regarding the dimeric fragment in the reduced lane (see the paragraph below Fig. R3.3 beginning with the text: “P10-11”).

5) Page 22: the end of the results section is missing

Our apologies for the formatting error, Figure 7 was somehow covering a small portion of text-- we have moved Figure 7 to reveal the last 1.5 lines of the paragraph in the PDF file.

Reviewer #2 (Remarks to the Author):

This is an interesting report on the conformational variability in the N-terminal domains of the archaeal AAA protein, PAN, a homolog of proteasomal ATPases from higher organisms. Using site directed mutagenesis to engineer proteins with strategically placed cysteine residues, the authors examined direct disulfide cross-linking patterns to show that the N-terminal coiled coil (CC) domains of PAN, which are needed to engage and unfold protein substrates, exist in different conformational states. They propose that the different conformational states are required for efficient unfolding of substrates and that the different conformations reflect an underlying asymmetry in PAN, and by extension other AAA proteins with similar functions, that is necessary for their activity. AAA proteins are ringed complexes that can be composed of identical or similar proteins. Nevertheless, studies of AAA proteins with diverse functions and from diverse organisms have given rise to an understanding that in executing their mechanical functions the subunits within the rings adopt different conformational states at any given time and that they cycle or otherwise transition between these states while hydrolyzing ATP and variously binding and letting go of polypeptide chains within the substrate protein. The current findings, if completely validated, would constitute a significant advance in helping to substantiate the current models of the mechanism and extend the notion of asymmetry to the linked N-terminal coiled-coil domain of PAN and similar enzymes.

The experiments performed in this study were carefully planned and elegantly executed. By placing cysteine residues at specific locations within the heptad repeats of the CC domain and quantifying the degree of spontaneous cross-linking, the authors showed that each of the three CC domains made up of helices from adjacent subunits have different contacts. Using spontaneous disulfide formation as a measure of closeness, the authors show that one CC has all five heptads in register, one CC has only its distal heptads in register, and that the third CC has the heptads shifted so that heptads 1,2, and 3 and in register with heptad 3,4 and 5 in its partner. While the logic of the experiments and their interpretation is unassailable, it remains that there is no direct chemical demonstration of disulfide bond formation and no chemical identification of the polypeptide chains in which the cross-linked residues reside. The conclusions of this study are too important to be based solely on correspondence between the calculated probability of forming a disulfide link and the apparent percentage of cross-linking observed. The cross-linked proteins need to be isolated, digested, and methods employed to identify chemically the residues forming the cross-link. Until that is done other explanations cannot be ruled out with complete confidence.

To address the reviewer's concern on the chemical identification of the crosslink, we have performed a mass spectrum analysis of oxidized PAN-M87C. In addition, we provide below a list of presented evidence that demonstrates the crosslink that we observe is indeed a disulfide bond between of the M87C residue.

MS analysis: MS analysis of the disulfide crosslinked peptides is not a trivial task for a variety of reasons (discussed here: Tsai et al. *Reviews in Analytical Chemistry*, 2013). Nevertheless we did attempt to confirm that the crosslinked PAN-M87C contained the expected disulfide

crosslink in the C1 conformation. Oxidized PAN-M87C was treated with Trypsin protease to generate peptide fragments of the CC domain, including the M87C residue. MS analysis by the “Scripps Center for Metabolomics” of this digestion was found to contain a peptide with a charge of +5 with a corresponding mass of 2886.456. This mass unambiguously matches the expected mass of an M87C-M87C disulfide crosslinked peptide. Note that this fragment has cleavage sites that were missed by trypsin (underlined), which is expected to occur more frequently when a disulfide bond impedes trypsin’s access to these cut sites (**Figure R2.1**):

No other expected trypsin digestion product or product multimer is expected to generate this peptide mass. These MS results therefore confirm that our oxidized PAN-M87C contains the expected disulfide crosslinked peptide, positively confirming the chemical identity of this crosslink. These results have been added to the manuscript text with a new accompanying supplemental figure (**Fig S3B**).

Evidence supporting M87C-M87C disulfide crosslink:

- 1) WT PAN does not form crosslinked dimers, but PAN-M87C does, demonstrating that only the mutated residue causes crosslinking.
- 2) PAN-M87C dimers do not form under reducing conditions but do form under oxidizing conditions. The only residues known to undergo such crosslinking under these conditions are Cysteines, demonstrating the crosslink is disulfide in origin.
- 3) Mass Spectrum analysis confirms the expected M87C-M87C disulfide crosslinked peptide.

We believe that these carefully controlled experiments unambiguously identify the nature of the PAN cross-linked dimers as presented in the manuscript.

The authors should also address the question of the degree to which allosteric effects upon cross-linking of the first CC in a hexamer might affect the ability to cross-linking other CCs. The presence of an allosteric influence would not invalidate the model and might even enrich it and it should be looked into.

This is a very interesting point, but not a trivial one to investigate experimentally. Indeed, it makes sense theoretically that limiting C1 conformational changes via crosslinking could also allosterically limit conformational rearrangements of C2 or C3. However, in these studies, PAN is 100% functional in all of its known activities when we crosslink the C1 conformation (M87C). Therefore, any allosteric limitation imposed on PAN by crosslinking the C1 conformation are likely to be unimportant for at least these basic functions of PAN (e.g. ATPase, protein unfolding, binding to the 20S, and inducing 20S gate-opening). To further test this hypothesis one would need to generate a PAN that could crosslink the C2 or C3 conformations without crosslinking the C1 conformation, but since PAN is a homohexamer and the C1 CC is fully in register and zipped, this would not be experimentally feasible using the approach we've developed here. The experiments could perhaps be done in the 26S ATPases, which are heterohexameric, but this would be outside the scope of this current study on PAN, though we agree that it would be very appropriate for a subsequent study.

Reviewer #3 (Remarks to the Author):

Overall, manuscript is clearly written and well-focused on the important topic of the fundamental aspects of proteasome function. The findings advance the field by providing an insight into the conformational states of the coiled coil domains of the archaeal PAN, a homolog of the Rpt1-6 AAA ATPases of 26S proteasomes.

Comments regarding experimentation:

P7 last two lines state [both oxidized and reduced forms of WT and M87C PAN formed hexamers via native-PAGE (Fig. S1A)] however, only the oxidized form of the PANs are displayed in Fig. S1A.

To address this we have modified this text to appropriately refer to Fig S1A as showing Native gels of the oxidized proteins:

“... the oxidized form of M87C PAN formed hexamers via native-PAGE (Fig S1A)...”

Fig. S1A: The data, used to argue the WT and variant PANs are all hexamers, are not convincing. The resolution of gel electrophoresis cannot be assessed, as only one protein band per lane is detected. Some type of control protein (e.g., PAN subcomplexes) is needed in the gel. If the PANs were similarly purified on a high-resolution gel filtration column calibrated with molecular mass standards, this type of information would be more convincing that the gels.

To address this we have included a gel (below) where we show molecular weight standards in the Native gel (**Fig R3.1**). Note that PAN runs on Native gels at a higher molecular weight than expected (~between the 480kDa and 720kDa “controls”, even though hexamers are ~300kDa), likely due to differences in tertiary/quaternary structure of PAN compared to the standards. In **Figure R3.1**, note that the PAN-M87C runs the same length in the same position as the WT control, indicating that PAN-M87C it adopts a similar size, charge, and tertiary structure as WT PAN. This figure and legend has now been added to the updated manuscript- **Figure S1**.

Figure R3.1:

Figure R3.1: Typical Native-PAGE of hexameric PAN (WT and M87C) alongside Native "standards". Note that although PAN hexamers are ~290 kDa, PAN hexamers migrate more slowly than expected in a Native gel (around ~600kDa), most likely due to the tertiary or quaternary structure the hexamers adopt. Therefore, when analyzing Native gels of PAN mutants, it is best practice to compare all mutants to a WT PAN control.

P9-10: disagree with statement that the evidence (provided at this point in the manuscript P9 In 10-11 which is a dose response with tetrathionate) indicates [that all cysteines within proximity to one another had successfully crosslinked, thus excluding possibility #1].

Any cysteines whose beta carbons are within 3.4-4.6 angstroms from one another with pseudobond angles of: $60^\circ < \theta_{ij}, \theta_{ji} < 180^\circ$, $0^\circ < |\theta_{ij} - \theta_{ji}| < 105^\circ$, will crosslink as explained in Careaga & Falke *Biophys J.* 1992. The fact that crosslinking saturates at ~33%, even as higher and higher concentrations of TT are added, indicates that one of the above conditions are not satisfied in at least ~67% of residues. We agree that if this were the only evidence provided that it would be less convincing. However, the results presented subsequently in figures 2D-2E further validate this statement with other approaches (and also rule out other explanations). Additionally, the experiments we later conduct to determine the conformations of C2 and C3 make more clear what conformations the coiled-coils adopt that indeed prevent them from crosslinking in the M87C mutant.

A control protein is needed that harbors a cysteine residue in similarly close-proximity to those of the PAN CC that can be fully cross-linked and fully reduced under the experimental conditions of this study to rule out possibility #1.

We think that the >95% crosslinking in **Figure 2E** satisfies this request for a control protein that crosslinks fully. Another control protein could have been useful as a positive control if we found ourselves unable to crosslink PAN fully, but we were able to do so. Furthermore, the major limitation of using a different positive control protein is that it is far less informative about PAN's ability to crosslink because: 1) it will not have similar cysteine rearrangements as PAN, so it would likely crosslink at different concentrations of tetrathionate, and 2) it would be almost impossible to pick a protein with the exact distances and angles that PAN's M87C cysteines have.

P10 Fig. 2E: the finding that in the presence of denaturant up to 96% of PAN proteins are dimers (vs. monomeric) is encouraging that the cross-linking conditions can capture any dimers that may be present. However, the overall level of PAN protein detected in the gel seems to have decreased upon increasing denaturant (e.g., the last lane on the right seems to have less protein than the first lane on the left in terms of total dimer + monomer). Please clarify. Is there any PAN protein retained in the well? The PAN could be x-linked into higher order structures that may be at low levels throughout the gel and, thus, contribute to the reduction in monomer levels. Are native cysteine residues found in the PAN? MJ1176 has 3 cysteines based on UniProtKB Q58576. As the PAN is denatured, how did you rule out that the native cysteine residues do not account for the reduced level of PAN monomers detected under oxidizing conditions?

Indeed, WT PAN does contain 3 WT cysteines (one between the OB and ATPase, and two near its C-terminus). However, these do not complicate analysis since the control, WT PAN, does NOT form dimers or higher MW species under the conditions we used, e.g. when SDS is added prior to removing the tetrothionate, see **Figure R3.2**. Therefore the WT cysteines do not affect dimerization status under these conditions. This is one reason we chose to use WT PAN as a control in all experiments done with cysteine mutants. Nevertheless, PAN-M87C does form some higher molecular weight species when denatured with SDS and analyzed via Native-PAGE. These are likely higher order structures that can form after the hexamer disassembly (See **Figure R1.4** above). Therefore, the increase in dimerization in this SDS dose response experiment can be attributed solely to the cysteines that were engineered into the CC domains. The slight loss of total protein at higher SDS (~20% by densitometry) most likely arises from the critical mini-spin column desalting step, which removes tetrothionate (see methods), as we do not observe any PAN retained in the well (see **Figure R3.2** below). In this particular experiment, some of the the SDS-induced high molecular weight forms (observed on Native gels; **Fig. R1.4**) were likely caught in the mini-spin columns thus slightly reducing the amount of recovered protein. Despite this, it is important to note that the absolute level of dimers actually increases in these higher SDS lanes, indicating a shift from monomers to dimers and not just a loss of monomers. Furthermore, the dimer ratio (dimer/total protein) increases in the lanes with higher concentrations of SDS to > 96% in M87C mutants. The quantification of this ratio was the intended purpose of this experiment.

Figure R3.2

P10 In 18-20: [while subcomplex II (F2) contained a 30-35 kDa fragment...] while unimportant to the conclusions of this manuscript - I cannot find these results which are described in the text.

Since subcomplex II (F2) was indeed unimportant to the conclusions it initially wasn't discussed further in the manuscript, we merely pointed out that we observed this peak on SEC. See **Figure R3.3** below for the SDS-PAGE and silver stain of the F2 peak (subcomplex II—the AAA+ ATPase portion of PAN). Note that SDS-PAGE analysis of the F2 peak show a primary band that is ~30-35 kDa in size, consistent with PAN subcomplex II monomers (e.g. the AAA+ ATPase domain fragment). Based on this feedback we have decided to include this figure in the supplement (**Fig S2**) for other that might be interested.

P10-11: the experiment associated with Fig. 3 is inconclusive in the absence of a PAN WT control. Panel C is particularly troubling. Based on the current evidence, an alternative explanation could be that the isolated CC-OB domain is tightly associated and cannot be fully denatured under SDS-PAGE gel conditions. The purity of the F3 subcomplex I of PAN M87C is also problematic. The proteolytic fragments observed in the SDS-PAGE gel in panel C, while indicated to be limited to high molecular mass fragments, are found to differ in profile between the oxidized and reduced states of the PAN M87C CC-OB domain and could also account for band indicated as a 'monomer' only based on migration in the gel. Thus, the oxidation state could have nothing to do with the migration of the M87C CC-OB domain fragment by SDS-PAGE. The claim that 100% of the CC-OB domain was cross-linked under oxidizing conditions (vs. the 33% observed for the full length) is hard to conclude – since the reduced state of the CC-OB is at 75% dimer vs. the full length at 0% dimer as observed by SDS-PAGE.

In order to compare the reduced and oxidized CC-OB domains in side-by-side lanes on a gel, we were required to run a non-reducing SDS-PAGE. Therefore, the presence of dimeric CC-OB domains in the reduced F3 sample is expected since the sample will oxidize to some extent in a non-reducing gel. The higher molecular weight species likely come from the fact that the F2 and F3 subcomplexes are not well resolved by SEC. Nevertheless, separation of the F2 and F3 subcomplexes is not necessary to demonstrate that the oxidized CC-OB domain was indeed fully dimerized, which this data shows. In fact, no monomer could be detected in the F3-Ox lane. The fact that the reduced CC-OB domain was, in contrast, not at least fully dimerized (despite the non-reducing gel), was also consistent with our model. Further, this result bolsters our

model since it shows that these crosslinks occur readily in the isolated CC-OB domain even in the absence of chemical oxidizing agents (unlike full-length PAN-M87C, which can only crosslink ~33%). This disulfide crosslinking analysis of the isolated CC-OB domain is exactly what one would expect to find based on the available crystal structures of the isolated CC-OB domain. Further biochemical experiments to demonstrate the crystal structure of the isolated CC-OB domain is correct seems unjustified. The only purpose of this experiment was to show that we could get 100% crosslinking of our M87C mutant if the ATPase domains were removed from the CC-OB domain, which this data clearly shows. To further address the reviewers concerns we have re-written the text to indicate that in-gel oxidation is likely, but it also leaves open other possibilities:

“Even a large fraction of the reduced form of the CC-OB fragment crosslinked, likely in the gel, due to the necessary non-reducing conditions (Fig 3C, F3-bottom of reduced lane).”

Statement regarding experimental reproducibility needs to be included for Fig. 3, Fig. S1-3 (please provide the number of experimental, biological, and technical replicates in the legend).

We have now indicated experimental reproducibility in the methods section *“Representative Images from 4 independent experiments are shown.”*

Please indicate p-values on each bar graph by * or some other standard notation (e.g., Fig. 4B states p-values for all points related to 87C were <0.001, yet the % crosslinking of M87C, M80C and M73C all appear to be around 33% and are argued so in the text.

80C and 73C are actually closer to ~40% crosslinked, and since these graphs are with $n > 10$, even this small ~7% increase was found to be statistically significant with $p < 0.001$. Therefore, in the text we state that: *“we observed a gradual increase in disulfide crosslinking formation from proximal to distal CC residues with the 59C mutation (5th heptad) resulting in $69 \pm 5.9\%$ crosslinking (59C-59C crosslink)”*.

p-values are particularly needed for Fig. 6C since the claim is that PAN WT and M87A differ in the rate of ATP hydrolysis in the absence of substrate.

We agree, and we have added statements of p-values in this figure.

P16: the authors discuss register shifts in terms of one or more ‘heptad’ (defined as a group or set of seven). This type of nomenclature is confusing to me – as a shift of one or more amino acid residue within the heptad is indicated by the data and within the figures. For example, a register shift of one heptad really seems to mean a register shift of one amino acid within the heptad.

We agree that this nomenclature can be confusing, however, this is the standard notation that is used to describe registry shifts in coiled-coils: for example, dynein undergoes a $\frac{1}{2}$ heptad

registry shift, which is actually only a registry shift of 1 amino acid on the hydrophobic layer (e.g. in one helix all 'a' residues shift to 'd' positions, and all 'd' residues shift to 'a' positions). All literature available on proteins with CCs that exhibit registry shifts use this nomenclature, so we therefore decided to be consistent with the literature and use this nomenclature—where a 2-heptad slide means that a 'd' residue in a heptad pairs with a 'd' residue 2 heptads away [e.g. the 1st heptad 'd' residue (87) pairs with the 3rd heptad 'd' residue (73), and the 2nd heptad 'd' residue (80) pairs with the 4th heptad 'd' residue (66)]

P20 In 8: disagree with term [must] - the data instead support the term [can]

We have changed this to read: “...*demonstrates that PAN's C1 CC can remain in-register and zipped...*”

P20 In 16: the authors do not provide data to demonstrate the substrate bound state of PAN.

GFPssrA is a substrate for PAN, and it is known that PAN binding to substrate increases PAN's ATPase activity, which has been shown in many publications. In addition, our results show that GFPssrA stimulates PAN's activity in Figure 6C, and since GFPssrA cannot stimulate PAN without binding to it, we think that it is appropriate to call this state “substrate-bound”. In addition, the concentration of GFP-ssrA that was used (3 μ M) was chosen because this is \sim 10X the K_d for GFPssrA-PAN interaction, which is therefore close to saturating.

P22 In 6-9: one scientific hurdle is that the 87C is common to locking both C1 (87C-87C) and C3 (87C-73C) states which makes the data less direct. Regarding locking PAN in different functional states - more details need to be provided in the methods section regarding the cross-linking conditions in terms of presence vs. absence of ATP or other nucleotide analog, temperature, buffer conditions, and range of protein concentration (only know less than or equal to 0.25 mg/ml). This information will help guide the reader regarding the functional state of PAN during the crosslinking.

To address this suggestion we have made changes to the text to clarify the buffer conditions in the methods as follows: “*All oxidation reactions were done in 50 mM Tris pH 7.5 + 5% glycerol (v/v), and were done in the absence of nucleotides, unless otherwise noted.*”

We indeed performed many experiments with various nucleotide states, and we did not find any significant changes in crosslinking with or without nucleotides, discussed in detail above in response to reviewer 1 (reference **Figure R1.1** and **Figure R1.2**). We believe that this result will be of interest to readers so we have included Figure R1.1 and R1.2 as new supplemental figures (**Fig S3A,C**) with a brief discussion in the manuscript text.

P24 In 6: disagree with statement [the present study provides the first direct structural evidence of this.] - the evidence is supportive but not direct or structural. In fact the statement P24 In 12-13 regarding a previous cryoEM study seems to provide direct structural evidence for asymmetry of CC domains of the AAA ATPases of the 26S proteasome.

For the sake of clarity the full sentence referenced above with regards to direct structural evidence is: *“Previous biochemical studies have suggested that PAN adopts asymmetrical conformations, but the present study provides the first direct structural evidence of this, at least for the CC domains.”*

We fully understand the reviewer’s point of view that crosslinking evidence may not seem like “structural” evidence, which is often visual in nature; However, the word “structural” does accurately describe what we’ve shown here with our disulfide crosslinking analysis—it is structural in that the two well defined cysteine residues must be adjacent to one another in space, within a 1.2 angstrom distance (as we explain and provide evidence for in the text) in order to undergo disulfide crosslinking. Thus we can make a conclusion about the structure of PAN—that PAN’s M87C residues on separate subunits are juxtaposed to one another in space – with a very high degree of resolution. This information puts limits on the 3D structure of PAN and so provides useful structural information. In addition, the only explanation for the C1, C2, and C3 conformations that we observe via engineered disulfide crosslinks is that there are direct structural (or conformational) differences in these identical CC domains. With regards to the referenced cryo-EM structures they are of the 19S, not PAN, and the resolutions of these structures are ~10-15 angstroms at the CC domains, so whether or not these structures can be used to conclude asymmetrical conformations in the CC domains, has not been demonstrated, though likely.

Figures:

Fig. 2. Need to clarify in Fig. 2 legend that Fig. S5 supports the quantitative analysis of PAN protein levels in the Fig. 2 gels and subsequent gels.

We have added the text to the end of the Figure 2 legend: “See Fig S7 for validation of quantitative SDS-PAGE analysis of PAN.” (note that the old figure S5 has now become the new figure S7)

Fig. 2B,2C-E, 3B-C, 4B, 5B-C, 7A, S2A, S2C and S5 are missing the molecular mass standards - yet the reader is expected to analyze Fig. 3C in terms of Mr and compare these molecular masses to the other gels that display the x-linked dimer and monomer. Providing only arrows that define each protein band in terms of dimer, monomer, etc. leads to assumptions that may not be justified. Indicating the migration of the Mr standards for the SDS-PAGE gels is important.

To address this concern we have added the requested molecular weight markers to Figure 2B, which is the first gel which shows monomers and dimers. Subsequent gels all have the appropriate controls (WT or M87C-reduced/oxidized) that were initially shown in Fig 2B for comparison. In addition, these gels contain only one protein (PAN) that is highly purified so the delineation of bands is not convoluted or challenging.

Fig. 3B, Gel filtration molecular weights are discussed in text but only presented in terms of elution volume in the figure. Furthermore, the methods section P31 In12 includes no definition of

the protein standards used for calibration and no listing of the dimensions of the Superose 12 column.

Based on this input, we are now including the molecular weight markers in the main text's **Figure 3C** (we have also inserted the figure below for your convenience). We have also updated the methods to show the specific Superose 12 column we used for this experiment: “*Superose 12 HR 10/30*”, which has approximately a 24ml capacity.

It's important to note (as we did in the text) that the method we used here to separate Subcomplex I and Subcomplex II was previously used in Zhang et al. *Mol. Cell* 2009 to obtain crystal structures of the CC-OB domain and AAA+ ATPase domain, respectively. As we mentioned previously (in our response regarding Figure S1), calculating the molecular weights from Native “standards” is not quantitative, since Native proteins migrate based on their molecular weight and charge, as well as their tertiary and quaternary structures. However, as a more appropriate reference, when purified full length PAN is run on this column it indeed elutes at the F1 fraction volume (~8-9mls). The F2 fraction from our gel filtration analysis had monomers of ~30-35 kDa on SDS-PAGE (**Fig. R3.3** and **new Figure S2**), which is consistent with the size expected for monomers from Subcomplex II (the AAA+ ATPase domain). The F3 fraction from our gel filtration analysis had monomers of ~7kDa in size, which is also consistent with the expected monomeric size of subcomplex I (the CC-OB domain). We believe these changes and clarifications address the reviewer's concerns.

Figure 3C (inserted from main manuscript)

Fig. 4 presented out-of-order (panel ACB) in the text.

We understand how this is a little confusing, but we think this is the best arrangement and manner to discuss this figure for the subsequent reasons: Fig 4C is a model of the C2 conformations that is presented after discussing how we arrived at the C2 conformation model.

In the first reference to 4C in the text, we refer to it only to make it easier to visualize what “partial unzipping” may look like, then refer to it at the end of this discussion as a complete model. To make the discussion more clear we have changed the text to say “(e.g. see Fig 4C)” instead of “(Fig 4C)”.

Fig. 6A-C, 7B-C, Fig. S3, Table S1-2: what do the numbers represent in terms of units of activity and Vmax? e.g., in panel 6A the ATPase rate is listed as 60 ATP per PAN per min? is this 60 moles of ATP per mole of PAN per min?

Yes, it can be considered moles of ATP per moles of PAN. The moles mathematically cancel out and so it is common practice to present it as “Substrate turnover per enzyme per minute” [in this case: ATP per PAN (hexamer) per minute].

Please define 100% in the footnote of Table S2.

In the figure legend we now state the following: “*Vmax values were calculated from curves in Figure S3 and values in Table S1. Vmax values were normalized to WT PAN controls and divided by the reduced form of the mutant*”.

Fig. S1: define the rate of LFP hydrolysis at 1-fold in terms of units/mg in the figure legend.

This particular experiment does not aim to make any conclusions about the specific activity of PAN, but rather makes the relative comparison between different forms of PAN (e.g. WT PAN functions similarly to PAN mutants with regards to 20S gate opening activity). Therefore, data in terms of units/mg are irrelevant. This practice is common in the field since no claims are made regarding the specific activity of the enzyme. The relative LFP hydrolysis rates are in relative fluorescence units (RFU) per time, and we therefore state in the text that “*The rate of LFP hydrolysis was calculated and fold stimulation of the 20S activity by PAN is shown. 20S alone control is considered 1-fold*”

Minor editing issues:

Some figures are out of order in terms of presentation and discussion within the manuscript (see specific comments)

-P1: briefly clarify early in the manuscript that each of the three-identical coiled coil (CC) domains discussed in the abstract are formed by the N-termini of two separate PANs (otherwise the reader is left wondering until P5 In 11-12)

We agree that it’s important to define this early, so we have addressed this below in P2 In 14+ comment.

-P1 In 13: define CC within abstract and text

We now introduce the term “CC” for the first time in the text and do not use the abbreviation in the abstract

-P1 In 12: [26S ATPases] need to clarify that the ATPases are not 26S

We have changed this to read: “26S proteasome’s ATPases”

-P2 In 10: the proteasome? which type 26S?

We have clarified this important point in the updated manuscript by now saying “26S *proteasome*”

P2 In 14+: [19S ATPase] need to clarify that the ATPases are not 19S

It is common in the mammalian proteasome literature to refer to the 20S, 19S and 26S as different compositions of the proteasome. These definitions are clearly stated at the start of the introduction (page 2 In 4-6). We realize this can be a bit confusing to someone outside of the field, but to maintain consistency within the field we think that we need to keep these nomenclatures as written. In this regard the “19S: is not a Svedberg coefficient but rather the name of the Regulatory Particle, as stated in the introductory paragraph. Alternatively, we could use the designation “RP” for regulatory particle, but this practice is most commonly used in the yeast literature regarding the proteasome, and thus would also add to confusion.

P2 In 14+: define what is meant by Rpt6/3 CC, Rpt1/2 CC and Rpt4/5 CC? see comment regarding the need to clarify that CC domains are each formed by the N-termini of two separate Rpt subunits earlier in the manuscript

We have now clarified this point by changing the sentence to the following:

“The CC domains are composed of α -helical extensions of the 19S’s AAA+ ATPase subunits (Rpt1-6) that dimerize to form 3 CCs (Rpt1/2, Rpt6/3, and Rpt4/5 CCs).”

-P2 In 19: define DUB and clarify which type of proteasome is meant by [the proteasome]

We agree that DUB is not common nomenclature outside of the proteasome field, so we have changed the text to read “...(e.g. the deubiquitinase USP14/Ubp6) at the 26S proteasome.”

P2 In 23-24: [This alone indicates that CC domains play fundamental roles in proteasome function] what evidence is this alone referring to? Am unclear which study alone is providing such striking evidence.

For clarity we’ve quoted the sentences here for reference:

“Thus, the CC domains physically connect substrate recruitment and ubiquitin processing to the unfolding machinery. This alone indicates that CC domains play fundamental roles in proteasome function.”

The word “This” in the second sentence is a conjunction connecting these two sentences as is common and acceptable use. The word “This” refers to the entire idea presented in the prior sentence. We see no way to make this more clear but are open to suggestions.

P2 last line: please define [their] in [their functional importance]

For convenience, here is the original sentence from the manuscript:

“Mutagenesis of the CC domains and CC peptide competition studies further corroborate their functional importance, since most perturbations to any CC domain render the 26S proteasome non-functional, leading to lethality”

The pronoun “their” refers to the CC domains, which were mentioned earlier in the sentence as well as immediately afterward in the sentence. We’ve tried to make this more clear in our new sentence (below), but feel that the term “CC domain” is used too many times (3x) in the new sentence, which is included in the current version of the manuscript:

“Mutagenesis of the CC domains and CC peptide competition studies further corroborate the functional importance of the CC domains, since most perturbations to any CC domain render the 26S proteasome non-functional, leading to lethality”

P3 In 1: switch between 26S vs. 26S proteasome is confusing

See our response to P2 In 14+: [19S ATPase] above.

P3 In 16-21: need to cite reference(s) to support the important points made in the first three sentences of this second paragraph

In the updated manuscript we now include a reference after each sentence in this section as requested.

P4 In 17: may not want to state so strongly - as the model substrates used in this study to demo PAN: proteasome function (GFPssrA and LFP-Amc) are not tagged with ubiquitin-like proteins or found in the archaeal cell

We have rephrased our sentence to provide as accurate of a summation of the literature as possible:

“Much like the 19S ATPases, some studies suggest that the CC domains from PAN are also involved in substrate binding, although it is thought that PAN can achieve this without the use of additional substrate receptors that are found in the 19S”

To our knowledge no study has ever shown that PAN has substrate receptors other than the CC domains, so we believe this statement to be factual and accurate. Even if other substrate recruiting factors for PAN are found in the future, the point that the CC domains bind to substrates that have been tested in the past is still true, even if evidence is limited to in vitro experiments.

P5 In 15-16 and 18: spell out species names upon first use and italicize throughout

In the updated manuscript we now spell out the first use and italicize species: “*Methanococcus jannaschii*” and “*Archaeoglobus fulgidus*”

Fig. S1A and other figures: please define 87C, 80C etc. in the figure legends (particularly the first figure legend). For example, that 87C refers to M87C, etc. Otherwise, the designation is easily confused with temperature in degrees Celsius.

Per the reviewers request we have modified the Figure legend of Figure 4 (the first place where this nomenclature appears) to include the following statement:

“In this and following figures we indicate the cysteine mutations by listing the residue number followed by a “C”, (e.g. residue M87 mutated to cysteine is denoted as “87C”).

Fig. S1A and text: need to define LFP (indicating that it is AMC linked)

In the S1 figure legend we now say:

“The stimulation of 20S activity (caused by PAN-induced 20S gate opening) was measured using saturating PAN and 2uM of a fluorescent reporter nonapeptide (LFP) with 10uM ATPγS and 20mM MgCl2 (see methods for details).”

In the new methods section, we now define the LFP nonapeptide as follows:

“In short, LFP contains a fluorescent reporter (MCA) at the N-terminus and a quenching group (DNP) at the C-terminus. Upon cleavage of the peptide by the 20S proteasome, MCA is released and an increase in fluorescence can be observed at ex/em: 325/393.

P9 reference to Fig. 6A-B is out of order and confusing

We agree that this reference to 6A-B in P9 can be confusing, since the reader may expect to read about it in the next paragraph. We have modified this in the revised manuscript to say “...and even hydrolyze ATP and unfold proteins (discussed in detail later, Fig. 6A & B)”.

P9 In 4-5: recommend modifying to state: PAN could exist in at least two different conformational populations including one that is 'crosslinkable' and another that is not...

The reviewer's request is that we add the words "at least two" to the above sentence, and while it is true that PAN can exist in more than two populations, our experimental design tests a binary distinction, either PAN subunits are cross-linked or not. Therefore, while there are many different conformations of PAN, all of them will either fall into 1 of 2 categories, cross-linked or not. Based on this we believe the sentence is written more accurately and is clearer without this phrase.

P18 20-21: [expected] used twice within one sentence

We have modified this sentence to read: "...verified by SDS-PAGE that the anticipated crosslinks were indeed present at the expected levels from the PAN..."

P21 In 3: please reword the sentence – since the rationale that [crosslinking the C1 conformation has no effect on activity]- does not support the point that [stabilizing the C3 conformation by crosslinking stabilizes a functional, but low activity state of PAN].

We agree that there was one logical step missing in this sentence, so in the updated manuscript we have changed the sentence to read: "*Since crosslinking the C1 conformation has no effect on activity, and since crosslinking C1+C3 lowers PAN's activity, these data indicate that stabilizing the C3 conformation by crosslinking stabilizes a functional, but low activity state of PAN.*" Stated in this way it becomes evident that the discussion of C1 crosslinking is very relevant in this sentence, as it serves as the control.

P28 In 21: T20S define

We have now defined "T20S" as 20S from *Thermoplasma acidophilum*.

P29 In 8+: define % in terms of v/v or w/v

We have now defined this as v/v in the case of glycerol, and w/v in the case of NaCl.

Throughout the manuscript - be sure to include space between the number and units - e.g, 100 ml vs. 100ml and avoid random capital letters (e.g., Pyruvate Kinase and Lactate dehydrogenase mid- sentence).

We have fixed these grammatical errors in the updated manuscript as requested.

P31 - Methods for partial proteolysis of PAN are not well defined. PAN was incubated with trypsin for 1 h in what buffer? How much of the quenching solution was added? What was the type of Superose 12 column (10/300 GL? this info will provide column dimensions), what type of

buffer was the column equilibrated in?

In the updated manuscript we have clarified each of these points in the methods as requested:

“0.4 mg/ml of PAN was mixed with increasing amounts of Trypsin (0-2 µg per 40 µl rxn), incubated for 1 hour at room temperature in reaction buffer (50 mM Tris pH 7.5, 5% v/v glycerol), and reactions were quenched with manufacturer recommendation amounts (1:100) of Halt Protease Inhibitor Cocktail (Thermo Scientific).”

“...injected onto a size exclusion column (Superose 12 HR 10/30, GE)”

References need a significant amount of formatting: ref. 4, 5,9,11,12,15,21,23,28,30,38,41,43,45,47,49 all need some type of correction e.g, use title case lettering, include volume numbers, italicize species names where appropriate, proofread for font errors such as ??, only include the article title once.

We have fixed the references in our updated manuscript as requested.

REVIEWERS' COMMENTS:

Reviewer #1 (Remarks to the Author):

In their revised version the authors have largely addressed my previous concerns, either experimentally or by adding clarifications to the text. Understanding the coupling of the different CCD conformations to the PAN activity state has been my major previous concern. I understand and accept that such control mechanism has not been characterized in detail for any AAA+ protein and that a respective analysis represents an independent study. I appreciate the text additions on possible regulatory modes by CCDs. Freezing PAN via the introduced crosslinking strategies in low and high activity states now offers a valuable tool to study PAN regulation in future studies.

The authors have addressed all further points; I therefore recommend publication of the manuscript.

Reviewer #2 (Remarks to the Author):

The authors still tend to use absolutes when qualified statements would serve as well and be more prudent.

Regarding the CC domains:

Abstract: "...coiled-coil domains SIMULTANEOUSLY adopt 3 different conformations.." Why not simply "...can be found in ..." The authors have not ruled out inductive effects. There could be many that are limited to just three by the cross-linking.

Abstract: "...while one coiled-coil's conformation never changes.." Stay closer to the observation: "...while one CC can be conformationally constrained, even..."

I. 194 "they MUST be conformationally asymmetric." Why must? How about "They adopt three non-identical conformations under these experimental conditions"

Abstract, I. 39: Awkward wording: the states weren't similarly proposed, similar states were proposed. Rewrite- "...mimic similar states proposed for the 26S..."

Reviewer #3 (Remarks to the Author):

The authors present a significant report on conformational states of the three identical coiled-coil domains (CCDs) of PAN, an archaeal AAA+ ATPase that functions with 20S proteasomes to degrade proteins in an energy-dependent manner. By use of site-specific disulfide crosslinking, the CCDs were found to simultaneously adopt 3 different states (denoted as C1, C2 and C3 – with C1 appearing static during the PAN ATPase cycle). Two models were developed from these findings: 1) the three states of the CCDs are not correlated with the conformational changes of the ATPase subunits needed to do work (i.e., protein unfolding, 20S gate opening and protein translocation into the 20S chamber) or 2) only the ATPase subunits associated in the C2 and C3 states are hydrolyzing ATP. Further studies are needed to understand why the C1 state remains static during ATPase cycle of binding of nucleotide (ATP, ADP and ATPgS), substrate and 20S proteasomes. Clarity on how the CCD conformations might control ATPase function remains to be determined.

POINT-BY-POINT RESPONSE TO REVIEWERS:

As requested by the editor, we have complied with the suggestions of reviewer #2:

Regarding the CC domains:

Abstract: "... coiled-coil domains SIMULTANEOUSLY adopt 3 different conformations..." Why not simply "can be found in..."

We have now removed the word "simultaneously" so the text now reads "coiled-coils can adopt three different conformations". This both satisfies the request of the reviewer, and helped to shorten the abstract to 150 words, as requested by the editor.

Abstract "... while one coiled-coil's conformation never changes.." stay closer to the observation "... while one CC can be conformationally constrained, even..."

As requested, we have changed this text to read: "we find that one coiled-coil can be conformationally constrained even while unfolding substrates"

l. 194 "they MUST be conformationally asymmetric. " Why must? How about "They adopt three non-identical conformations under these experimental conditions"

As requested, we have changed this text to read: "they adopt three non-identical conformations under these experimental conditions"

Abstract, l. 39: Awkward wording: the states weren't similarly proposed, similar states were proposed. Rewrite- "mimic similar states proposed for the 26S..."

As requested, we have changed this text to read "... mimics similar states proposed for the 26S..."